# Numerical Experiments on Performance Comparisons of Conical Type Direct-Acting Relief Valve—With or without Conical Angle in Valve Element and Valve Seat

**Huiyong Liu [1,\*] and Qing Zhao [2]**

1 Department of Mechanical Design, School of Mechanical Engineering, Guizhou University, Guiyang 550025, China
2 Department of Water Resources and Hydropower Engineering, College of Civil Engineering, Guizhou University, Guiyang 550025, China
* Correspondence: heartext@163.com

**Abstract:** This paper conducts numerical experiments on performance comparisons of CTDARV—with or without conical angle in the valve element and valve seat. The working principles of three kinds of CTDARV are introduced. The simulation models of three kinds of CTDARV are established by utilizing AMESIM. Numerical experiments on CTDARV, with or without a conical angle in the valve element and the valve seat, are conducted and the performance comparisons of three kinds of CTDARV are obtained. The results show that: (1) When all parameters of VED, VSD, VEM, SS, CAVE&CAVS, and OD have the same value, respectively, CA-VE has the highest stable pressure, CA-VE&VS has the highest stable displacement, CA-VS has the lowest stable pressure, and CA-VE has the lowest stable displacement. The stable pressure of CA-VE is significantly higher than that of CA-VS and CA-VE&VS. The stable displacement of CA-VE&VS is significantly higher than that of CA-VE and CA-VS, and the stable displacement of CA-VE and CA-VS has little difference. (2) With the increase of VED from 13 mm to 16 mm, the stable pressure of CA-VE remains constant, while that of CA-VS and CA-VE&VS both decreases. As the VSD increases from 3 mm to 6 mm, the stable pressure of CA-VE and CA-VE&VS decreases, and that of CA-VE decreases significantly. With the increase of VEM from 0.01 kg to 0.04 kg, the stable pressure of CA-VE, CA-VS, and CA-VE&VS remains unchanged. With the increase of SS from 5 N/mm to 20 N/mm, the stable pressure of CA-VE, CA-VS and CA-VE&VS increases. With the increase of CAVE&CAVS from 15 degrees to 60 degrees, the stable pressure of CA-VE, CA-VS, and CA-VE&VS decreases. With the OD increase from 0.8 mm to 1.4 mm, the stable pressure of CA-VE, CA-VS and CA-VE&VS remains unchanged. (3) With the increase of VED from 13 mm to 16 mm, the stable displacement of CA-VE will not change, while that of CA-VS and CA-VE&VS will increase. As the VSD increases from 3 mm to 6 mm, the stable displacement of CA-VE increases, while that of CA-VE&VS decreases. When VSD is 4 mm–6 mm, the stable displacement of CA-VS remains unchanged. With the increase of VEM from 0.01 kg to 0.04 kg, the stable displacement of CA-VE, CA-VS and CA-VE&VS remains unchanged. As SS increases from 5 N/mm to 20 N/mm, the stable displacement of CA-VE, CA-VS, and CA-VE&VS decreases. As CAVE&CAVS increases from 15 degrees to 60 degrees, the stable displacement of CA-VE, CA-VS, and CA-VE&VS decreases. With the OD increasing from 0.8 mm to 1.4 mm, the stable displacement of CA-VE, CA-VS, and CA-VE&VS remains unchanged. (4) With the increase of VED from 13 mm to 16 mm, the velocity of CA-VE remains unchanged, while that of CA-VS and CA-VE&VS increases. As the VSD increases from 4 mm to 6 mm, the velocity of CA-VS remains unchanged, while that of CA-VE and CA-VE&VS decreases. With the increase of VEM from 0.01 kg to 0.04 kg, the velocity oscillation of CA-VE gradually increases, and the velocity of CA-VS and CA-VE&VS has little change. As SS increases from 5 N/mm to 20 N/mm, the velocity of CA-VE increases, while that of CA-VS and CA-VE&VS decreases. When CAVE&CAVS is 15 degrees and 30 degrees, the velocity of CA-VE is lower than that of CA-VS and CA-VE&VS. With the OD increasing from 0.8 mm to 1.4 mm, the velocity oscillation of CA-VE increases gradually, and the velocity of CA-VS and CA-VE&VS changes little.





**Keywords:** CTDARV; performance comparisons; AMESIM

## 1. Introduction

The cone valve type direct-acting relief valve (CTDARV) is widely used in low-pressure and small-flow systems because of its advantages of good sealing performance, fast response, strong anti-pollution ability, etc. Generally, the CTDARV can be divided into three categories according to whether the valve element or the valve seat has a conical angle: with a conical angle in the valve element but without a conical angle in the valve seat (CA-VE), without a conical angle in the valve element but with a conical angle in the valve seat (CA-VS), with a conical angle in both the valve core and the valve seat (CA-VE&VS). In the hydraulic systems of industrial equipment, these three types of CTDARVs have been widely used. However, under the same working medium, geometric parameters, and external working conditions, what are the response characteristics of the three types of CTDARVs? How is the performance comparison of these three types of CTDARVs? Can these three types of CTDARVs be directly replaced with each other? These problems have always plagued designers and on-site staff for a long time. Therefore, it is of great significance to carry out numerical experiments on performance comparisons of these three types of CTDARVs.

During recent decades, there have been lots of papers about the CTDARV. Yuan, et al. [1] conducted a numerical study on the cavitating flow phenomenon inside poppet valves with two valve seat structures aiming to examine the flow mechanisms underlying varying cavitation phenomena at different openness. Min et al. [2] measured the discharge coefficient of different pilot-stage poppet valves and used Fluid-Structure Interaction to analyze the influence of the orifice-submerged jet on the discharge coefficient. Hao et al. [3] established a prediction model of flow force on the cone and cavitation for the poppet valve by using CFD combined with the Zwart–Gerber–Belamri cavitation model and investigated the effects of three poppet valve configurations and their parameters on the flow force and cavitation intensity in valves. Yuan, et al. [4] performed a three-dimensional simulation considering the compressibility of each constituent phase to clarify the governing mechanisms under the cavitating flow inside two water poppet valves. Fornaciari et al. [5] dealt with experimental tests and numerical simulations (3D and 0D fluid-dynamic modeling) of a conical poppet pressure relief valve with flow force compensation. Min et al. [6] studied the unstable vibration of the poppet and cavitation using visual experiments. Filo et al. [7] used CFD to analyze an innovative directional control valve consisting of four poppet seat valves and two electromagnets enclosed inside a single body. Wang et al. [8] established the overflow rate of a poppet relief valve and the dynamics characteristics of a poppet relief valve and derived the stability conditions of a poppet relief valve system using the Routh–Hurmitz method. Gomez et al. [9] developed a computational model based on the finite volume method to characterize the flow at the interior of the valve while it is moving. Chiavola, et al. [10] experimentally characterized the behavior of a conical poppet valve and explored the effect of the feeding conditions on the flow characteristics of the valve. Lei et al. [11] established the flow model of the poppet valve orifice with a novel function of flow discharge coefficient and established the dynamic model including the aforementioned flow model of the poppet valve by considering the fluid forces caused by the valve body motion and the flowrate variation. Liu et al. [12] presented the regulation methods of flow forces acting on the main poppet in a large flow load control valve. Ji et al. [13] studied the dynamics of axial vibration and lateral vibration coexisting in a poppet valve. Han et al. [14] presented a numerical investigation into the flow force and cavitation characteristics inside water hydraulic poppet valves. Bo [15] presented the fluid-solid coupling model for the optimization of a poppet valve, and analyzed the actual flow performance of the poppet valve. Wang et al. [16] proposed a hydraulic poppet valve dynamic numerical model based on the CFD and dynamic mesh technique for obtaining the transient characteristics of hy-

draulic poppet valve during opening process. Yi et al. [17] investigated experimentally the interactions between the poppet vibration characteristics and cavitation property in relief valves with the unconfined poppet. Min et al. [18] presented a fluid-structure interaction modeling of poppet fluid, and simulated the response of poppet under the action of step and periodic excitation signal, and analyzed dynamic characteristics of viscous force and hydraulic force on poppet surface. Zheng and Quan [19] optimized the structure of the poppet valve based on the internal flow, and studied the flow-force on poppet valve in the case of the converging flow using CFD. Shi and Chen [20] simulated the inside flow field of the hydraulic poppet valve by the dynamic mesh technology based on the physical numerical modeling of the hydraulic poppet valve. Shi [21] established and meshed the 3-D model of the poppet valve, and simulated the inner flow field of the poppet valve at different cone diameter and different opening positions. Yi et al. [22] obtained the frequency spectrum of the squeal noise by analyzing the sampling data from the accelerometer mounted on the valve body. Chen and Bao [23] developed a simulation model to study the static characteristic of a regulator valve with large flow capacity. Luan et al. [24] established a numerical model of the hydraulic poppet valve using the CFD software in order to obtain the transient distribution of flow and valve core stress of the hydraulic poppet valve during opening process. Rundo and Altare [25] analyzed different methods for the evaluation of the flow forces in conical poppet valves, and analyzed three different poppet angles and two flow directions. Guo et al. [26] developed a 6 DOF model of a three-dimensional poppet valve to stimulate the transient flow field in the opening process by means of computational fluid dynamics software-Fluent.

However, the existing literatures have not yet compared the performance of CTDARV--with or without conical angle in valve element and valve seat. This paper carries out numerical experiments on three kinds of CTDARV, hoping to obtain the performance comparisons of three kinds of CTDARV, thus providing theoretical basis for design, manufacture and use of three kinds of CTDARV.

The rest of this paper is organized as follows. In Section 2, the working principles of three kinds of CTDARV are introduced. In Section 3, the simulation models of three kinds of CTDARV are established by utilizing AMESIM. The numerical experiments on three kinds of CTDARV are conducted and the performance comparisons of three kinds of CTDARV are obtained in Section 4. Finally, some conclusions are drawn in Section 5.

## 2. Working Principles of Three Kinds of CTDARV

Figure 1 is the structural diagram of three kinds of CTDARV. The typical feature of Figure 1a is that there is a conical angle in valve element and no conical angle in the valve seat (CA-VE). The typical feature of Figure 1b is that there is a conical angle in the valve seat and no conical angle in the valve element (CA-VS). The typical feature of Figure 1c is that there is a conical angle in the valve element and valve seat (CA-VE&VS). The working principle of the CTDARV can be described in two steps.

(1)    Before movement of the valve element, as shown in Figure 1a–c.

At this step, the valve element is subject to the joint action of the following forces: the upward hydraulic force on the valve element $F_1$, the downward hydraulic force on the valve element $F_2$, the mass force $G$, the spring force $F_s$, and the friction force $f$.

For CA-VE, CA-VS, CA-VE&VS, at this step, the equations of $F_1$, $F_2$, $G$, $F_s$, $f$ are (1), (2), (3), respectively.

$$\begin{cases} F_1 = \frac{\pi}{4}d^2 p_1 + \frac{\pi}{4}(D^2 - d^2)p_2 \\ F_2 = \frac{\pi}{4}D^2 p_2 \\ G = mg \\ F_s = k(x_0 + x) \\ f = \mu A \frac{v}{h} \end{cases} \tag{1}$$

$$\begin{cases} F_1 = \frac{\pi}{4}d^2 p_1 \\ F_2 = \frac{\pi}{4}D^2 p_2 \\ G = mg \\ F_s = k(x_0 + x) \\ f = \mu A \frac{v}{h} \end{cases} \tag{2}$$

$$\begin{cases} F_1 = \frac{\pi}{4}d^2 p_1 + \frac{\pi}{4}(D^2 - (d + l\sin\alpha)^2)p_2 \\ F_2 = \frac{\pi}{4}D^2 p_2 \\ G = mg \\ F_s = k(x_0 + x) \\ f = \mu A \frac{v}{h} \end{cases} \tag{3}$$

where, $D, d$ are the valve element diameter and the damping hole diameter; $p_1, p_2$ are the pressure in the front chamber of the valve element and in the spring chamber; $m$ is the mass of the valve element; $g$ is the gravity acceleration; $k$ is the spring stiffness; $x_0, x$ are the pre-compression and deformation of the spring, respectively; $\mu$ is the dynamic viscosity; $A$ is the surface area between the valve element and the hydraulic oil; $v$ is the velocity difference between the valve element and the valve body; and $h$ is the distance between the valve element and the valve body.

It can be seen from Figure 1b that $p_2$ is connected to the outlet, so $p_2 = 0$, and $F_2 = 0$. When $p_1$ is low, thus $F_1 \leq F_s + f + G$, the valve element does not move, and the valve core and valve seat contact.

(2)　After movement of valve element, as shown in Figure 1d–f

At this step, the valve element is subject to the joint action of the following forces: the upward hydraulic force on the valve element $F_1$, the downward hydraulic force on the valve element $F_2$, the mass force $G$, the spring force $F_s$, the friction force $f$, and the flow force $F_f$. The equations of $F_1$, $F_2$, $G$, $F_s$, $f$, $F_f$ are as follows:

For CA-VE, CA-VS, CA-VE&VS, at this step, the equations of $F_1$, $F_2$, $G$, $F_s$, $f$, $F_f$ are (4), (5), (6), respectively.

$$\begin{cases} F_1 = \frac{\pi}{4}d^2 p_1 + \frac{\pi}{4}(D^2 - d^2)p_2 \\ F_2 = \frac{\pi}{4}D^2 p_2 \\ G = mg \\ F_s = k(x_0 + x) \\ f = \mu A \frac{v}{h} \\ F_f = \rho q(\beta_2 v_2 \cos(\alpha) - \beta_1 v_1) \end{cases} \tag{4}$$

$$\begin{cases} F_1 = \frac{\pi}{4}d^2 p_1 \\ F_2 = \frac{\pi}{4}D^2 p_2 \\ G = mg \\ F_s = k(x_0 + x) \\ f = \mu A \frac{v}{h} \\ F_f = \rho q(\beta_2 v_2 \cos(\alpha) - \beta_1 v_1) \end{cases} \tag{5}$$

$$\begin{cases} F_1 = \frac{\pi}{4}d^2 p_1 + \frac{\pi}{4}(D^2 - (d + l\sin\alpha)^2)p_2 \\ F_2 = \frac{\pi}{4}D^2 p_2 \\ G = mg \\ F_s = k(x_0 + x) \\ f = \mu A \frac{v}{h} \\ F_f = \rho q(\beta_2 v_2 \cos(\alpha) - \beta_1 v_1) \end{cases} \tag{6}$$

where, $\rho$ is the density of the hydraulic oil; $q$ is the flowrate of oil flowing through the valve element and the valve seat; $\beta_1, \beta_2$ are the momentum correction coefficient at the inlet and outlet of the valve element; $v_1, v_2$ are the velocity at the inlet and outlet of the valve element; and $\alpha$ is the half angle of the valve element. The description of other symbols is the same as mentioned above.

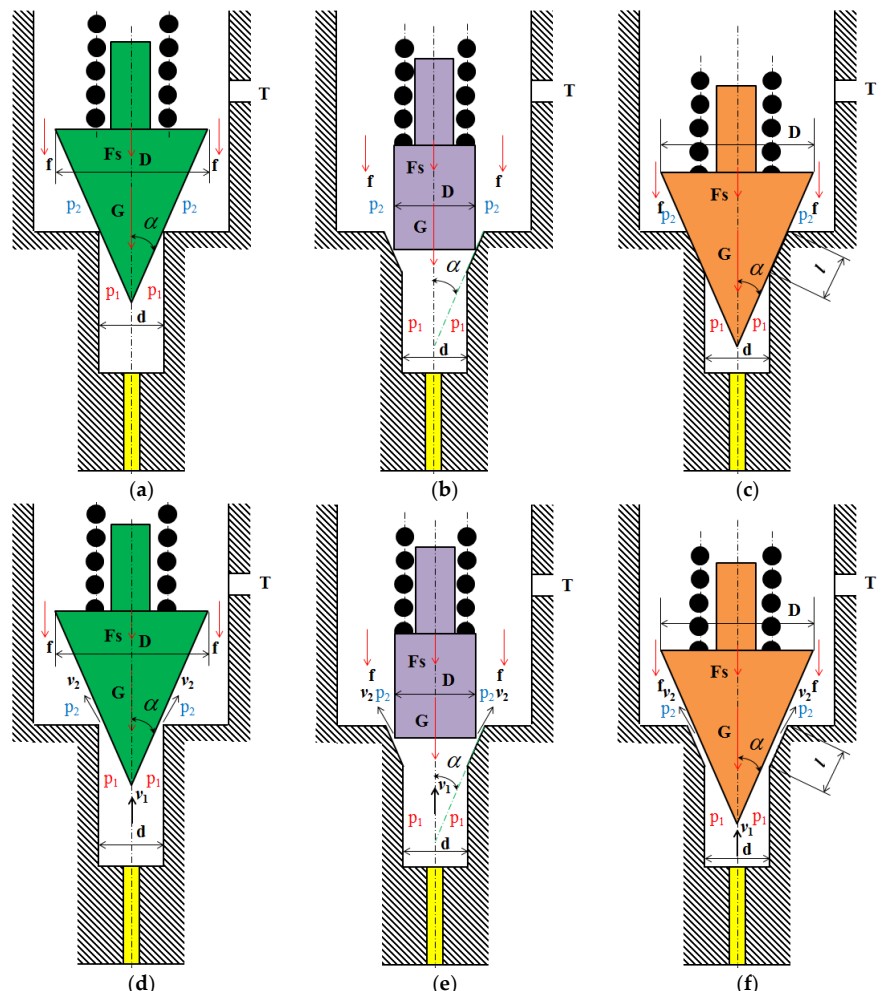

**Figure 1.** The structural diagram of three kinds of CTDARV. (**a**) CA-VE: Stage 1; (**b**) CA-VS: Stage 1; (**c**) CA-VE&VS: Stage 1; (**d**) CA-VE: Stage 2; (**e**) CA-VS: Stage 2; (**f**) CA-VE&VS: Stage 2.

It can be seen from Figure 1b that $p_2$ is connected to the outlet, so $p_2 = 0$, and $F_2 = 0$. When $p_1$ is high, thus $F_1 > F_s + f + G$, the valve element moves upwards, and the valve core and valve seat are separated, as shown in Figure 1c. When $F_1 = F_s + f + F_f + G$, the valve element moves upward for a certain distance and then maintains the stable state, and the high-pressure oil overflows to the tank through the outlet.

## 3. AMESIM Simulation Models of Three Kinds of CTDARV

In recent years, some famous software including AMESim, MATLAB, EASY5, HOPSAN have been widely employed in hydraulic system simulation. The biggest advantage of AMESim, compared with other software, i.e., MATLAB, is that it can free users from time-consuming programming and tedious mathematical modeling.

AMESim is an advanced modeling and simulation environment built upon the basis of a bond graph, composed of many libraries such as Hydraulic, Mechanical, Control, etc. Users can build complex system models just in AMESim to conduct simulation and analysis. HCD is the library for designing hydraulic components in AMESim, which can help users to establish almost all hydraulic components to study the performance of hydraulic components.

Based on the working principle of the three kinds of CTDARV, the AMESIM simulation models of the three kinds of CTDARV with/without orifice are built by utilizing the Hydraulic library, the HCD library, and the Mechanical library, as shown in Figure 2. In these models, the thick solid line represents the oil pressure action surface, and the arrow

represents the pressure action direction. The following basic assumptions were made when building the AMESIM simulation models:

(1) The operating temperature and ambient temperature remain constant;
(2) The physical and chemical properties of the working medium remain constant;
(3) No geometric shape error and assembly error in all parts;
(4) No internal and external leakage;
(5) No deformation in all parts during operation.

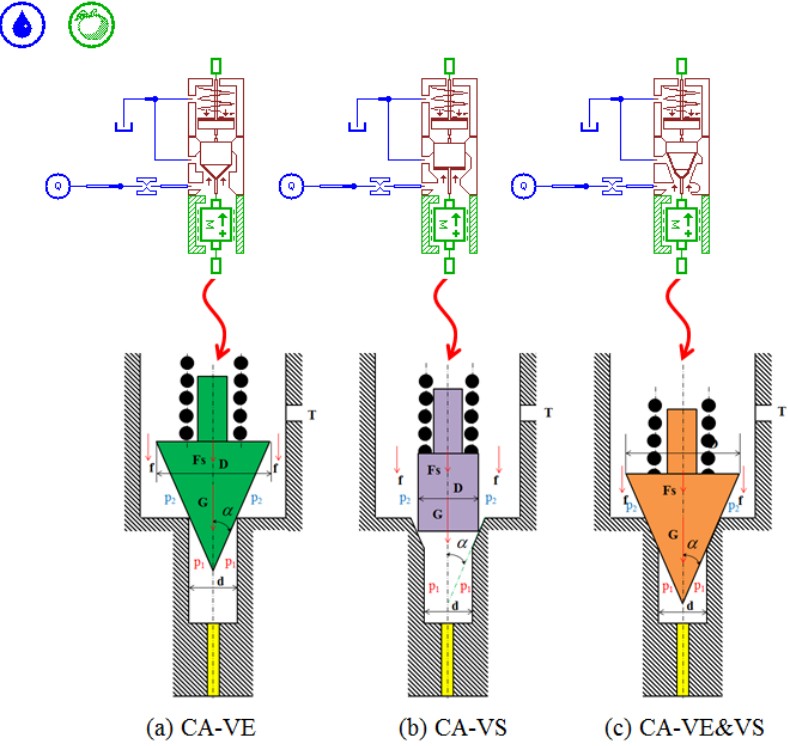

**Figure 2.** AMESIM simulation models of the three kinds of CTDARV.

## 4. Results and Discussion

The basic parameters are: the working temperature $T = 40\,^\circ\text{C}$, the density of hydraulic oil $\rho = 850\ \text{kg/m}^3$, the bulk modulus of hydraulic oil $K = 17000$ bar, the absolute viscosity of hydraulic oil $\nu = 51$ cP, and the flowrate of hydraulic oil $Q = 15\ \text{L/min}$. The simulation parameters are shown in Table 1, including the valve element diameter (VED), valve seat diameter (VSD), valve element mass (VEM), spring stiffness (SS), conical angle of valve element and valve seat (CAVE&CAVS), and the orifice diameter (OD). By setting the parameters in the AMESIM simulation models of the three kinds of CTDARV, performance comparisons can be obtained.

**Table 1.** Simulation parameters.

| VED | VSD | VEM | SS | CAVE & CAVS | OD |
|---|---|---|---|---|---|
| (mm) | (mm) | (kg) | (N/mm) | (°) | (mm) |
| 13, 14, 15, 16 | 5 | 0.01 | 10 | 45 | 1 |
| 15 | 3, 4, 5, 6 | 0.01 | 10 | 45 | 1 |
| 15 | 5 | 0.01, 0.02, 0.03, 0.04 | 10 | 45 | 1 |
| 15 | 5 | 0.01 | 5, 10, 15, 20 | 45 | 1 |
| 15 | 5 | 0.01 | 10 | 15, 30, 45, 60 | 1 |
| 15 | 5 | 0.01 | 10 | 45 | 0.8, 1.0, 1.2, 1.4 |

### 4.1. Pressure Performance Comparisons of the Three Kinds of CTDARV

4.1.1. Effect of VED on Pressure Response

Figure 3 shows the effect of VED (13 mm–16 mm) on the pressure response of the three kinds of CTDARV. It can be seen from Figure 3 that when the VED is 13 mm–16 mm, the three kinds of CTDARV will eventually reach their respective stable pressure, of which CA-VE has the highest stable pressure, and CA-VS has the lowest stable pressure. The stable pressure of CA-VE is significantly higher than that of CA-VS and CA-VE&VS. The pressure of CA-VE fluctuates, but the number of fluctuations is small, while the pressure of CA-VS and CA-VE&VS does not fluctuate. The pressure of CA-VE reaches its stable pressure at 0.02 s, while CA-VS and CA-VE&VS reach their respective stable pressure at 0.001 s. With the increase of VED from 13 mm to 16 mm, the stable pressure of CA-VE will not change, while the stable pressure of CA-VS and CA-VE&VS will decrease. Specifically, when the VED is 13 mm, 14 mm, 15 mm, and 16 mm, the stable pressure of CA-VE is 68.92 bar; the stable pressure of CA-VS is 8.24 bar, 7.07 bar, 6.14 bar, and 5.28 bar; and the stable pressure of CA-VE&VS is 12.51 bar, 11.21 bar, 10.08 bar, and 9.11 bar, respectively.

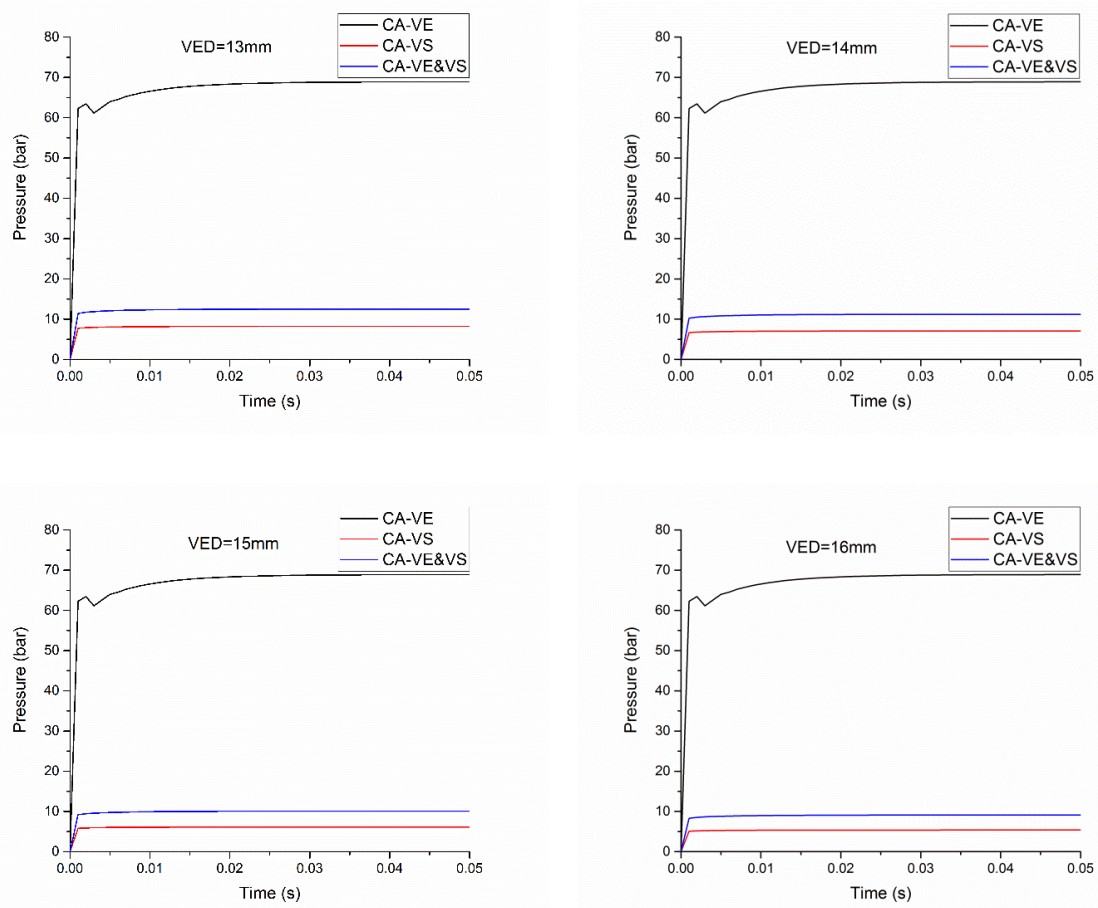

**Figure 3.** The pressure response of the three kinds of CTDARV: VED = 13 mm–16 mm.

4.1.2. Effect of VSD on Pressure Response

Figure 4 shows the effect of VSD (3 mm–6 mm) on the pressure response of the three kinds of CTDARV. It can be seen from Figure 4 that when VSD is 3 mm–6 mm, respectively, the three kinds of CTDARV will eventually reach their respective stable pressure, of which CA-VE has the highest stable pressure, and CA-VS has the lowest stable pressure. The stable pressure of CA-VE is significantly higher than that of CA-VS and CA-VE&VS. The smaller the VSD, the longer the pressure fluctuation times of CA-VE, and the longer the time required to reach the stable pressure. When VSD is 6 mm, the pressure of CA-VE does not fluctuate. When VSD is 3 mm–6 mm, the pressure of CA-VS and CA-VE&VS do

not fluctuate. When the VSD is 3 mm, the pressure of CA-VS reaches the stable pressure of 0.02 s, while when the VSD is 4 mm–6 mm, the pressure of CA-VS reaches the stable pressure at 0.001 s. When VSD is 3 mm–6 mm, the pressure of CA-VE&VS reaches stable pressure at 0.001 s. With the increase of VSD from 3 mm to 6 mm, the stable pressure of CA-VE and CA-VE&VS decreases, and the stable pressure of CA-VE decreases significantly. The stable pressure of CA-VS is larger when the VSD is 3 mm, while it keeps the same small value when the VSD is 4 mm–6 mm. Specifically, when VSD is 3 mm, 4 mm, 5 mm, and 6 mm, the stable pressure of CA-VE is 229.08 bar, 115.24 bar, 68.92 bar, 45.73 bar, respectively; that of CA-VS is 10.83 bar, 6.14 bar, 6.14 bar, and 6.14 bar, respectively; and that of the CA-VE&VS is 10.84 bar, 10.28 bar, 10.08 bar, 9.96 bar, respectively.

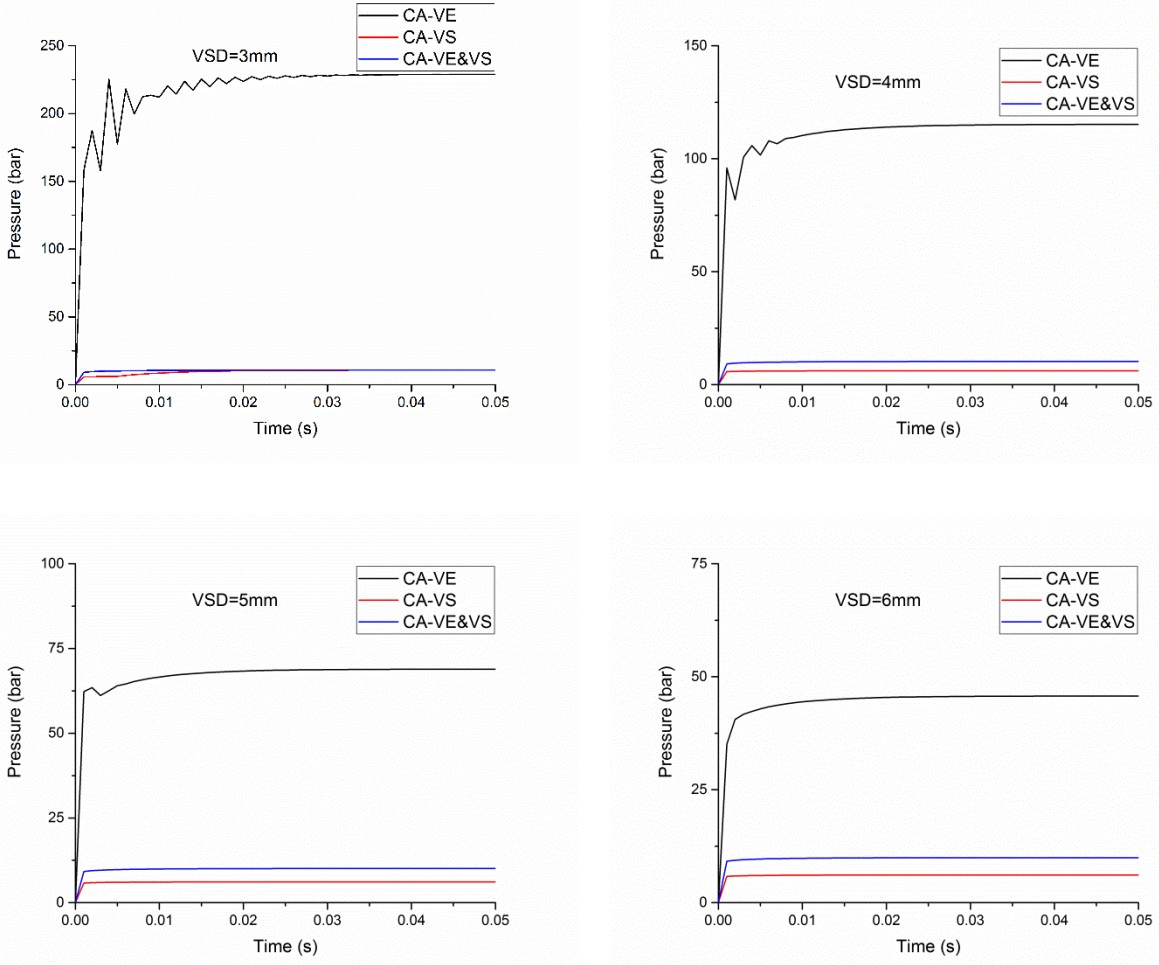

**Figure 4.** The pressure response of the three kinds of CTDARV: VSD = 3 mm–6 mm.

### 4.1.3. Effect of VEM on Pressure Response

Figure 5 shows the effect of VEM (0.01 kg–0.04 kg) on the pressure response of the three kinds of CTDARV. It can be seen from Figure 5 that when the VEM is 0.01 kg–0.04 kg, the three kinds of CTDARV will eventually reach their respective stable pressure, of which CA-VE has the highest stable pressure, and CA-VS has the lowest stable pressure. The stable pressure of CA-VE is significantly higher than that of CA-VS and CA-VE&VS. The greater the VEM, the longer the pressure fluctuation times of CA-VE, and the longer the time required to reach the stable pressure. When the VEM is 0.01 kg and 0.02 kg, the pressure of CA-VE&VS reaches the stable pressure in 0.001 s. When VEM is 0.03 kg and 0.04 kg, the pressure of CA-VE&VS reaches stable pressure at about 0.003 s and 0.005 s, respectively. When VEM is 0.01 kg–0.04 kg, the pressure of CA-VS reaches stable pressure at 0.001 s. With the increase of VEM from 0.01 kg to 0.04 kg, the stable pressures of CA-VE,

CA-VS and CA-VE&VS remain unchanged. Specifically, when the VEM is 0.01 kg, 0.02 kg, 0.03 kg, and 0.04 kg, the stable pressure of CA-VE, CA-VS, and CA-VE&VS is 68.92 bar, 6.14 bar, and 10.08 bar, respectively.

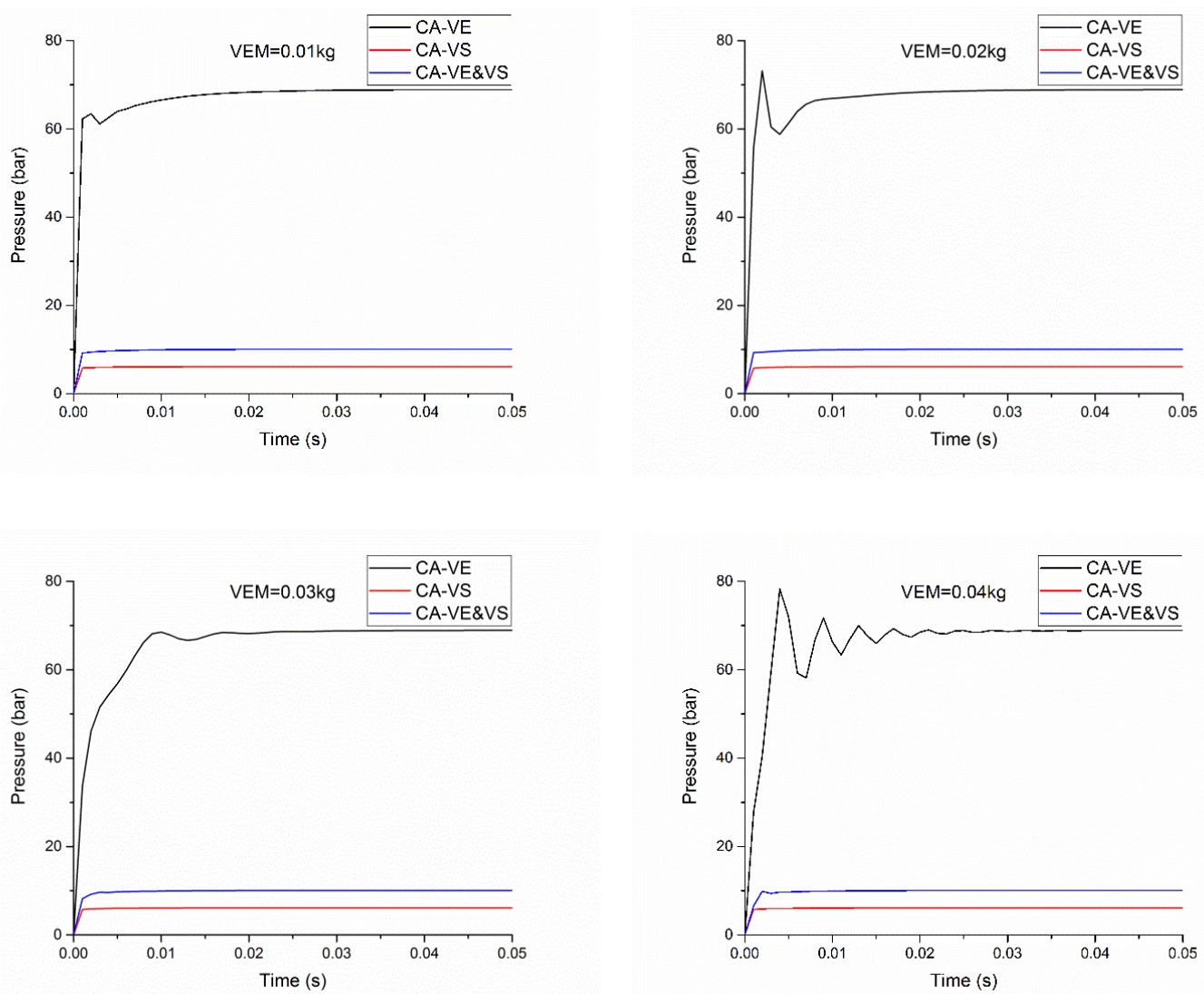

**Figure 5.** The pressure response of the three kinds of CTDARV: VEM = 0.01 kg–0.04 kg.

### 4.1.4. Effect of SS on Pressure Response

Figure 6 shows the effect of SS (5 N/mm–20 N/mm) on the pressure response of the three kinds of CTDARV. It can be seen from Figure 6 that when SS is 5 N/mm–20 N/mm, the three kinds of CTDARV will eventually reach their respective stable pressure, of which CA-VE has the highest stable pressure, and CA-VS has the lowest stable pressure. The stable pressure of CA-VE is significantly higher than that of CA-VS and CA-VE&VS. When SS is 5 N/mm–20 N/mm, the pressure of CA-VE fluctuates less, and the pressure of CA-VS and CA-VE&VS does not fluctuate. When SS is 5 N/mm–20 N/mm, respectively, the pressure of CA-VS and CA-VE&VS reaches stable pressure at 0.001 s. With the increase of SS from 5 N/mm to 20 N/mm, the stable pressure of CA-VE, CA-VS and CA-VE&VS increases, but the increase is very small. Specifically, when SS is 5 N/mm, 10 N/mm, 15 N/mm, and 20 N/mm, the stable pressure of CA-VE is 68.17 bar, 68.92 bar, 69.65 bar and 70.38 bar, respectively; and that of CA-VS is 6.06 bar, 6.14 bar, 6.22 bar, and 6.30 bar, respectively; and that of CA-VE&VS is 9.76 bar, 10.08 bar, 10.40 bar, and 10.70 bar, respectively.

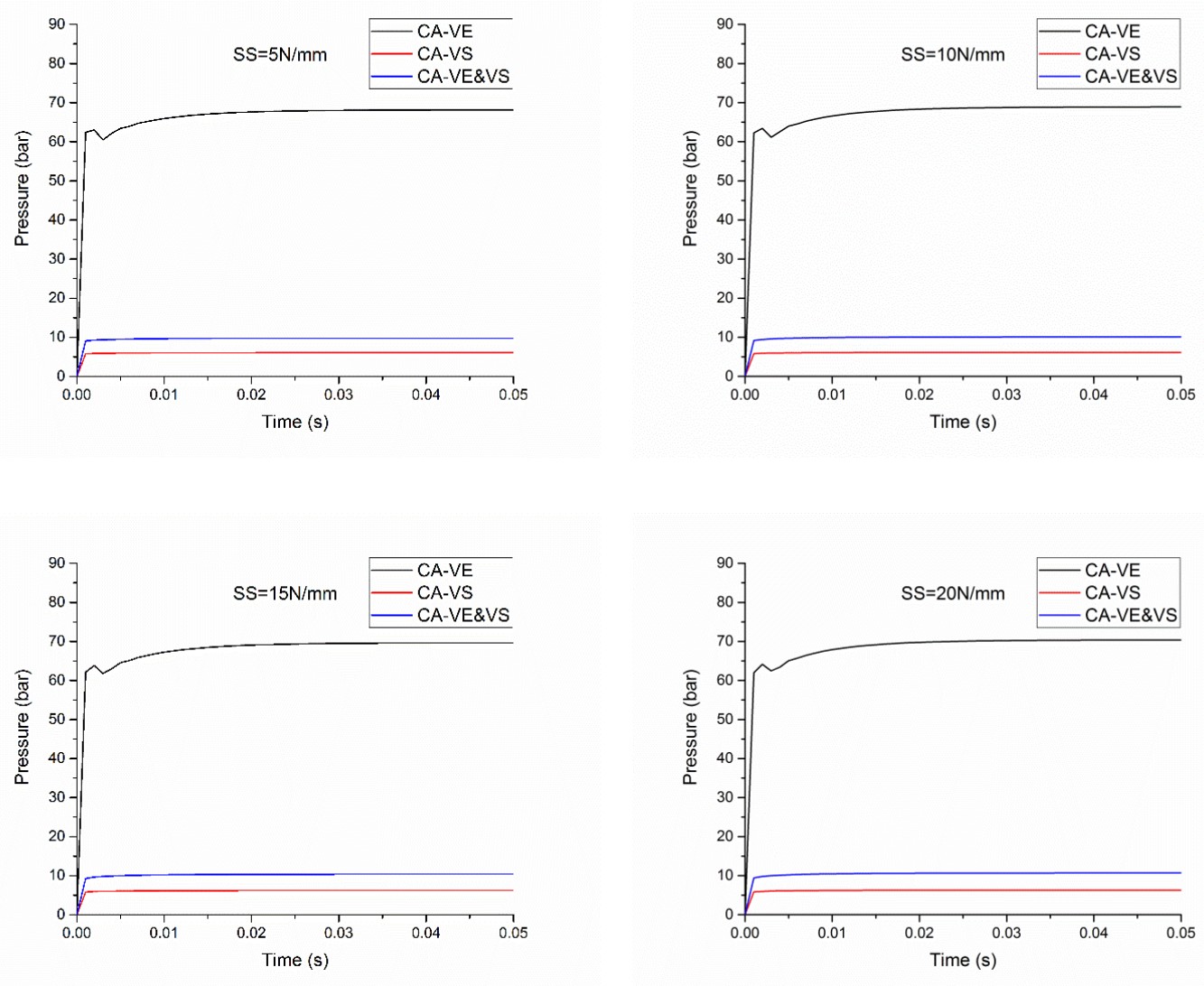

**Figure 6.** The pressure response of the three kinds of CTDARV: SS = 5 N/mm–20 N/mm.

### 4.1.5. Effect of CAVE&CAVS on Pressure Response

Figure 7 shows the effect of CAVE&CAVS (15 degrees–60 degrees) on the pressure response of the three kinds of CTDARV. It can be clearly seen from Figure 7 that when CAVE&CAVS are 15 degrees 60 degrees, respectively, the three kinds of CTDARV will eventually reach their respective stable pressure, of which CA-VE has the highest stable pressure and CA-VS has the lowest stable pressure. The stable pressure of CA-VE is significantly higher than that of CA-VS and CA-VE&VS. When CAVE&CAVS is 15 degree–60 degrees, the pressure of CA-VS and CA-VE&VS does not fluctuate. When CAVE&CAVS is 30 degrees and 45 degrees, respectively, the pressure of CA-VE does not fluctuate. When CAVE&CAVS is 15 degrees and 60 degrees, respectively, the pressure of CA-VE fluctuates, but the number of fluctuations is small. When CAVE&CAVS is 15 degrees 60 degrees, respectively, the pressure of CA-VS and CA-VE&VS reaches stable pressure at 0.001 s. With the increase of CAVE&CAVS from 15 degrees to 60 degrees, the stable pressure of CA-VE, CA-VS, and CA-VE&VS decreases, but the decrease is not significant. Specifically, when CAVE&CAVS are 15 degrees, 30 degrees, 45 degrees, and 60 degrees, the stable pressure of CA-VE is 78.65 bar, 73.86 bar, 68.92 bar, and 63.35 bar, respectively; the stable pressure of CA-VS is 6.53 bar, 6.28 bar, 6.14 bar, and 6.02 bar, respectively; and the stable pressure of CA-VE&VS is 11.40 bar, 10.50 bar, 10.08 bar, and 9.77 bar, respectively.

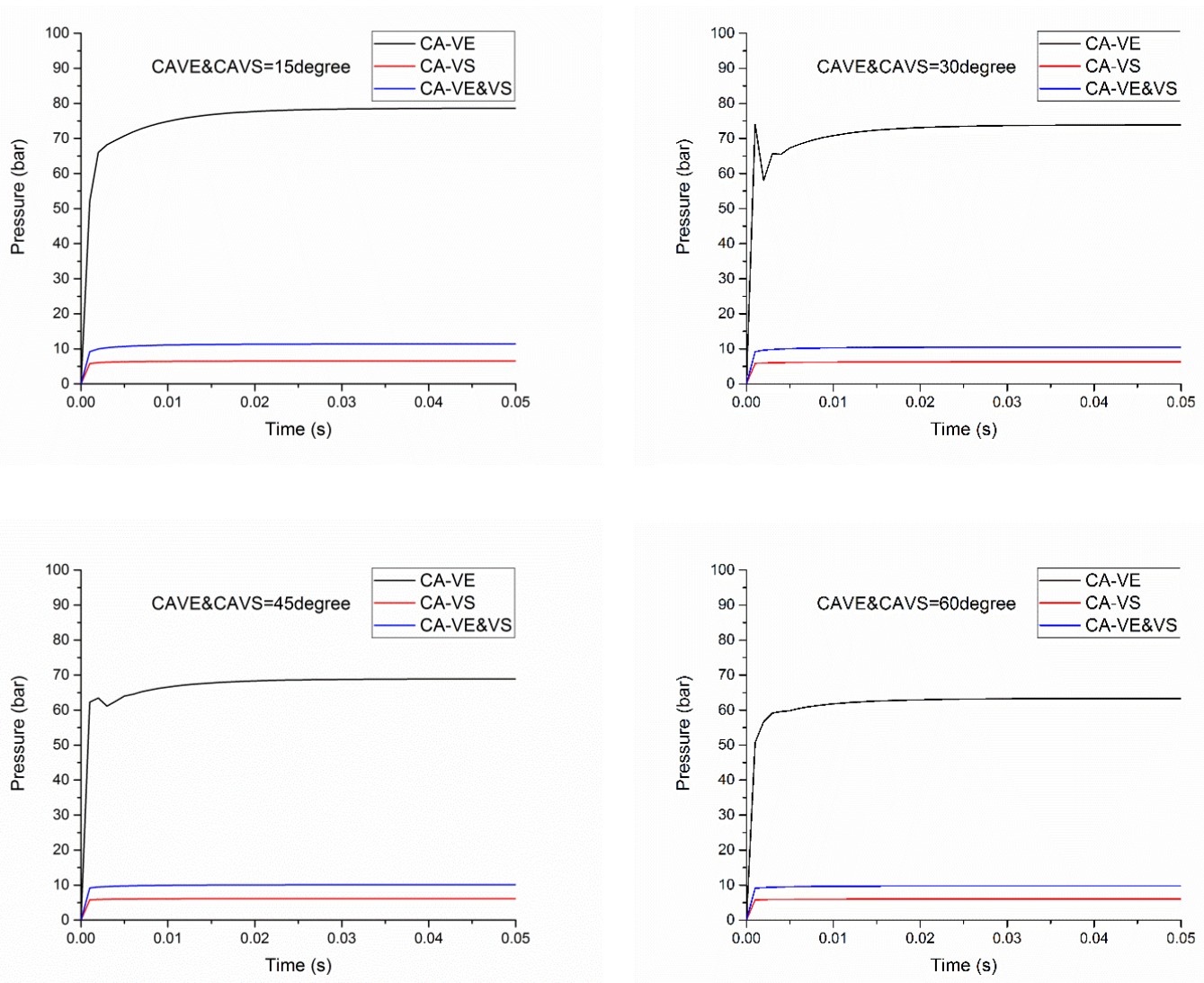

**Figure 7.** The pressure response of the three kinds of CTDARV: CAVE&CAVS = 15 degrees–60 degrees.

### 4.1.6. Effect of OD on Pressure Response

Figure 8 shows the effect of OD (0.8 mm–1.4 mm) on the pressure response of the three kinds of CTDARV. It can be clearly seen from Figure 8 that when the OD is 0.8 mm–1.4 mm, respectively, the three kinds of CTDARV will eventually reach their respective stable pressure, of which CA-VE has the highest stable pressure and CA-VS has the lowest stable pressure. The stable pressure of CA-VE is significantly higher than that of CA-VS and CA-VE&VS. When OD is 0.8 mm–1.4 mm, the pressure of CA-VS and CA-VE&VS does not fluctuate. When OD is 0.8 mm, the pressure of CA-VE does not fluctuate. When the OD is 1.0 mm–1.4 mm, respectively, the pressure of CA-VE fluctuates, but the number of fluctuations is small. The greater the OD, the greater the pressure fluctuation of CA-VE. When the OD is 0.8 mm–1.4 mm, respectively, the pressure of CA-VS and CA-VE&VS reaches stable pressure at 0.001 s. With the increase of OD from 0.8 mm to 1.4 mm, the stable pressures of CA-VE, CA-VS, and CA-VE&VS remain unchanged. Specifically, when the OD is 0.8 mm, 1.0 mm, 1.2 mm and 1.4 mm, the stable pressure of CA-VE is 68.92 bar, the stable pressure of CA-VS is 6.14 bar, and the stable pressure of CA-VE&VS is 10.08 bar, respectively.

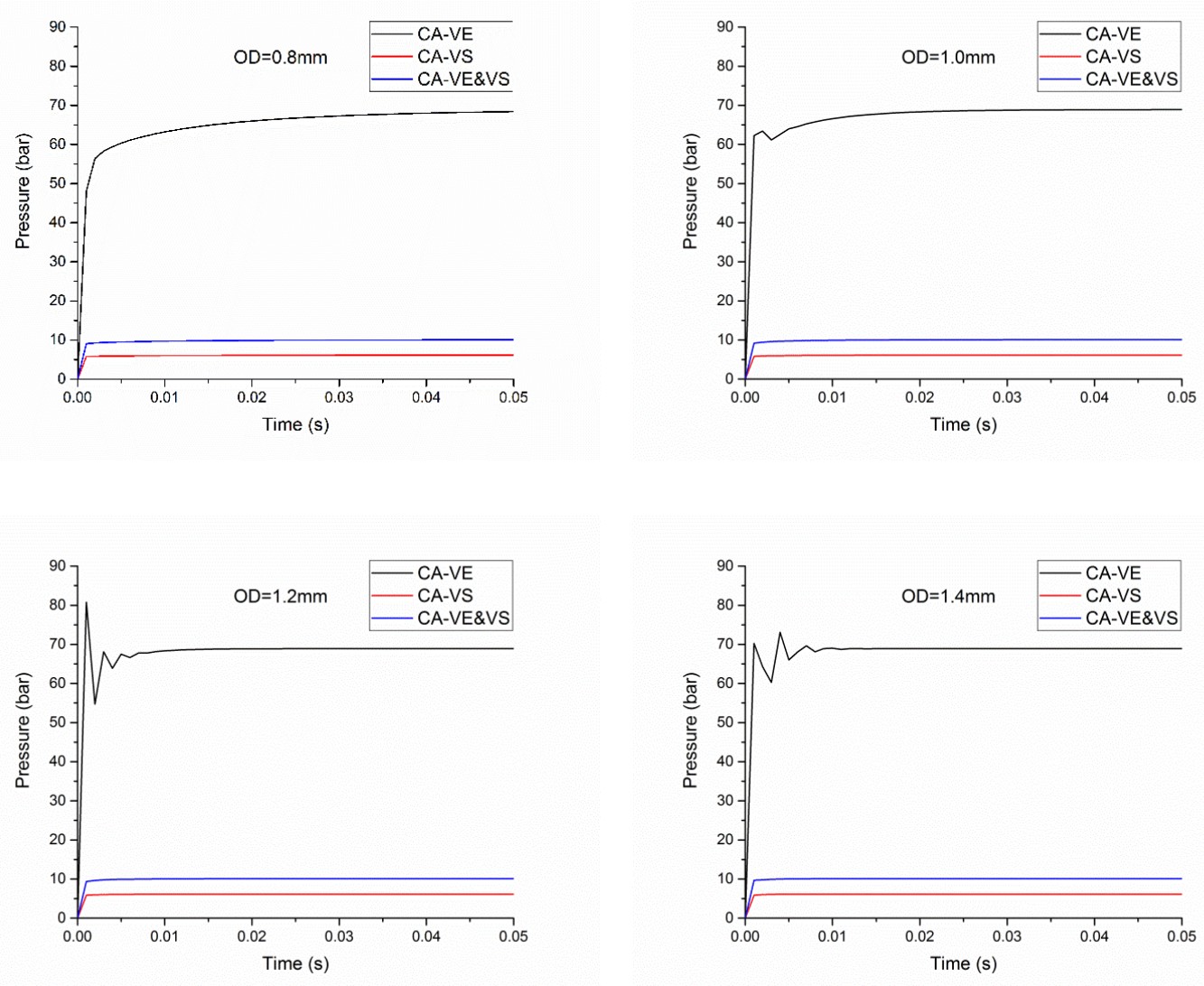

**Figure 8.** The pressure response of the three kinds of CTDARV: OD = 0.8 mm–1.4 mm.

*4.2. Flowrate Performance Comparisons of the Three Kinds of CTDARV*

4.2.1. Effect of VED on Flowrate Response

Figure 9 shows the effect of VED (13 mm–16 mm) on the flowrate response of the three kinds of CTDARV. It can be clearly seen from Figure 9 that when the VED is 13 mm–16 mm, the three kinds of CTDARV will eventually reach a stable flowrate of 15 L/min, and all have similar flowrate curves. It can be seen from the partial enlarged diagram that the CA-VS has relatively large flowrate and the CA-VE has relatively small flowrate in the initial stage before 0.001 s. At 0.001 s, the flowrate of CA-VE, CA-VS and CA-VE&VS decreased with the increase of VED. Specifically, when the VED is 13 mm, 14 mm, 15 mm, and 16 mm, the flowrate of CA-VE is 3.5102 L/min, 3.2074 L/min, 2.8822 L/min, and 2.5346 L/min, respectively; the flowrate of CA-VS is 5.7207 L/min, 5.65757 L/min, 5.5783 L/min, and 5.4804 L/min, respectively; and the flowrate of CA-VE&VS is 5.0839 L/min, 4.8622 L/min, 4.5983 L/min, and 4.2873 L/min, respectively.

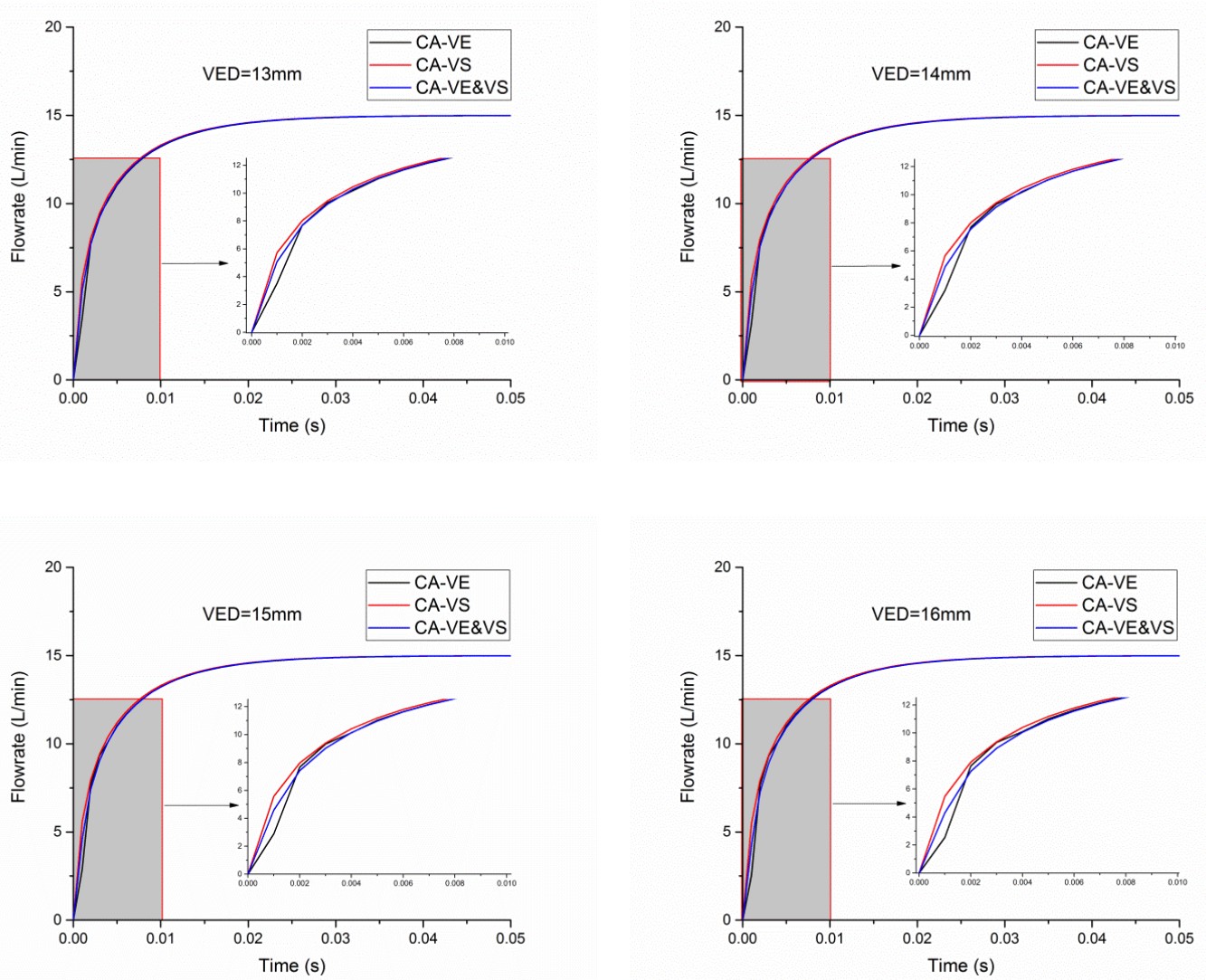

**Figure 9.** The flowrate response of the three kinds of CTDARV: VED = 13 mm–16 mm.

#### 4.2.2. Effect of VSD on Flowrate Response

Figure 10 shows the effect of VSD (3 mm–6 mm) on the flowrate response of the three kinds of CTDARV. It can be clearly seen from Figure 10 that when the VED is 13 mm–16 mm, the three kinds of CTDARV will eventually reach a stable flowrate of 15 L/min. It is worth noting that when the VSD is 3 mm, the flowrate of CA-VS oscillates violently before 0.015 s. When the VSD is 4 mm–6 mm, the flowrate of CA-VS does not fluctuate. When VSD increases from 3 mm to 6 mm, the flowrate fluctuation of CA-VE decreases gradually, while the flowrate of CA-VE&VS does not fluctuate. When VSD is 5 mm and 6 mm, it can be seen from the local enlarged view that CA-VS has relatively large flowrate and CA-VE has relatively small flowrate in the initial stage before 0.001 s. At 0.001 s, with the VSD increasing from 5 mm to 6 mm, the flowrate of CA-VE and CA-VE&VS increased, and the flowrate of CA-VS remained unchanged. Specifically, when VSD is 5 mm and 6 mm, the flowrate of CA-VE is 2.8822 L/min and 4.6088 L/min, respectively; the flowrate of CA-VS is 5.5783 L/min, and 5.5783 L/min, respectively; and the flowrate of CA-VE&VS is 4.5983 L/min, and 4.8690 L/min, respectively.

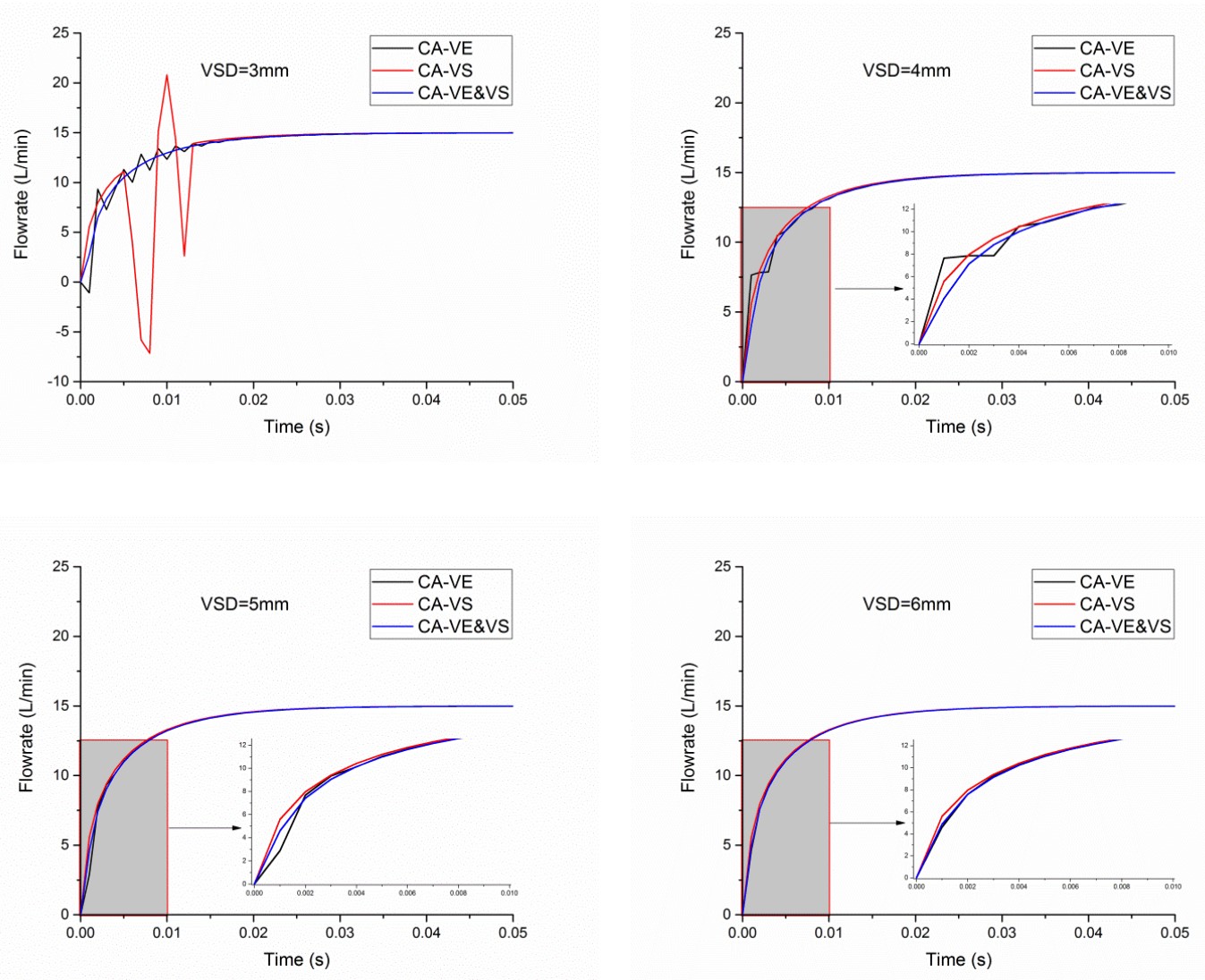

**Figure 10.** The flowrate response of the three kinds of CTDARV: VSD = 3 mm–6 mm.

### 4.2.3. Effect of VEM on Flowrate Response

Figure 11 shows the effect of VEM (0.01 kg–0.04 kg) on the flowrate response of the three kinds of CTDARV. It can be clearly seen from Figure 11 that when the VEM is 0.01 kg–0.04 kg, the three kinds of CTDARV will eventually reach a stable flow of 15 L/min. When VEM is 0.01 kg, the flow of CA-VE does not fluctuate. When VEM increases from 0.02 kg to 0.04 kg, the flow fluctuation of CA-VE increases gradually. When VEM increased from 0.01 kg to 0.04 kg, the flow of CA-VS and CA-VE&VS did not fluctuate. When VEM is 0.01 kg, it can be seen from the local enlarged view that CA-VS has relatively large flow rate and CA-VE has relatively small flow rate in the initial stage before 0.001 s. At 0.001 s, the flow of CA-VE is 2.8822 L/min, the flow of CA-VS is 5.5783 L/min, and the flow of CA-VE&VS is 4.5983 L/min.

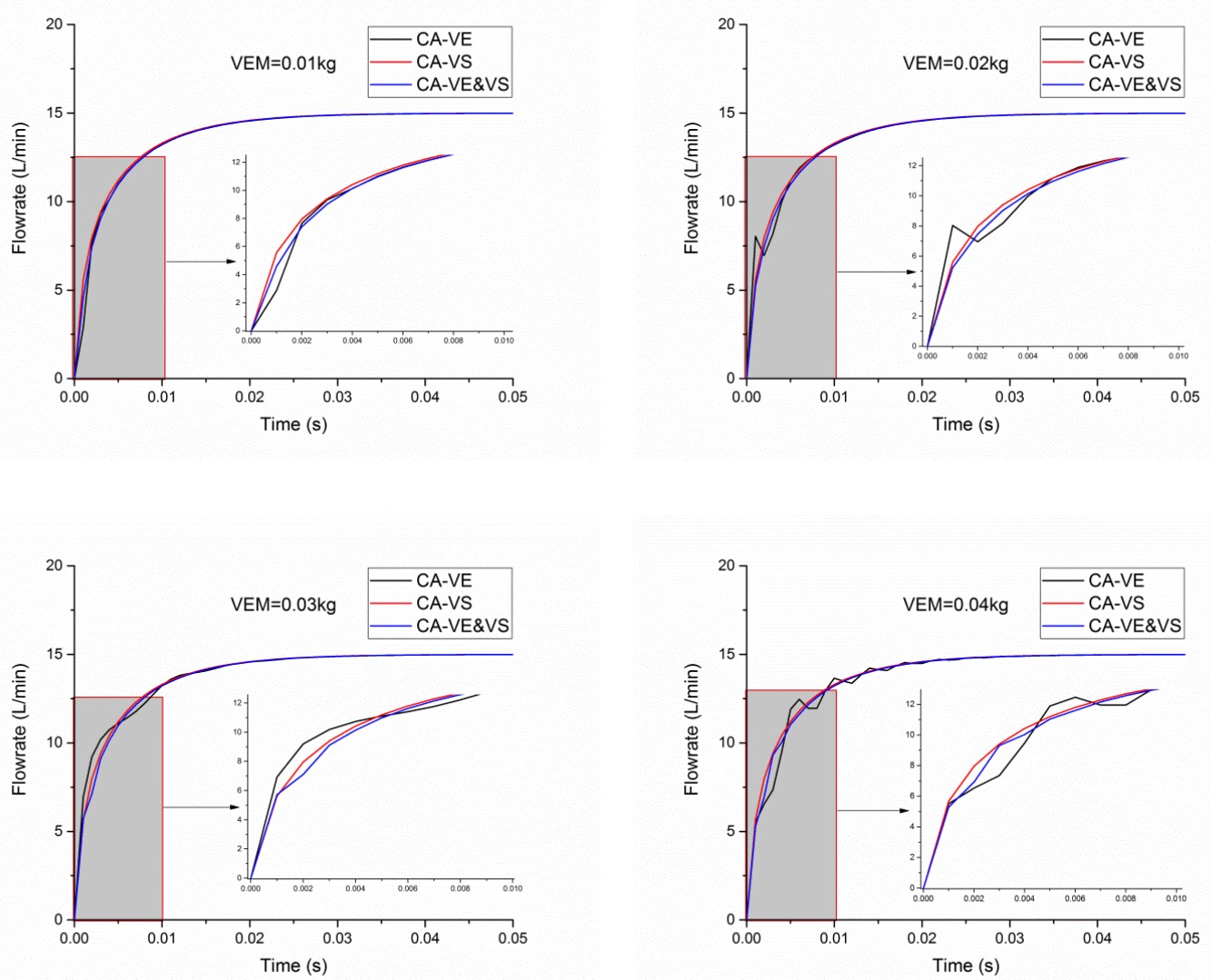

**Figure 11.** The flowrate response of the three kinds of CTDARV: VEM = 0.01 kg–0.04 kg.

### 4.2.4. Effect of SS on Flowrate Response

Figure 12 shows the effect of SS (5 N/mm–20 N/mm) on the flowrate response of the three kinds of CTDARV. It can be clearly seen from Figure 12 that when SS is 5 N/mm–20 N/mm, the three kinds of CTDARV will eventually reach a stable flowrate of 15 L/min, and all have similar flowrate curves. It can be seen from the partial enlarged diagram that the CA-VS has relatively large flowrate and the CA-VE has relatively small flowrate in the initial stage before 0.001 s. At 0.001 s, with SS increasing from 5 N/mm to 20 N/mm, the flowrate of CA-VE decreases, while the flowrate of CA-VS and CA-VE&VS increases, but the decrease and increase are not significant. Specifically, when SS is 5 N/mm, 10 N/mm, 15 N/mm and 20 N/mm, the flowrate of CA-VE is 2.9047 L/min, 2.8822 L/min, 2.8620 L/min, and 2.8440 L/min, respectively; the flowrate of CA-VS is 5.5754 L/min, 5.5783 L/min, 5.5813 L/min, and 5.5842 L/min, respectively; and the flowrate of CA-VE&VS is 4.5796 L/min, 4.5983 L/min, 4.6159 L/min, and 4.6327 L/min, respectively.

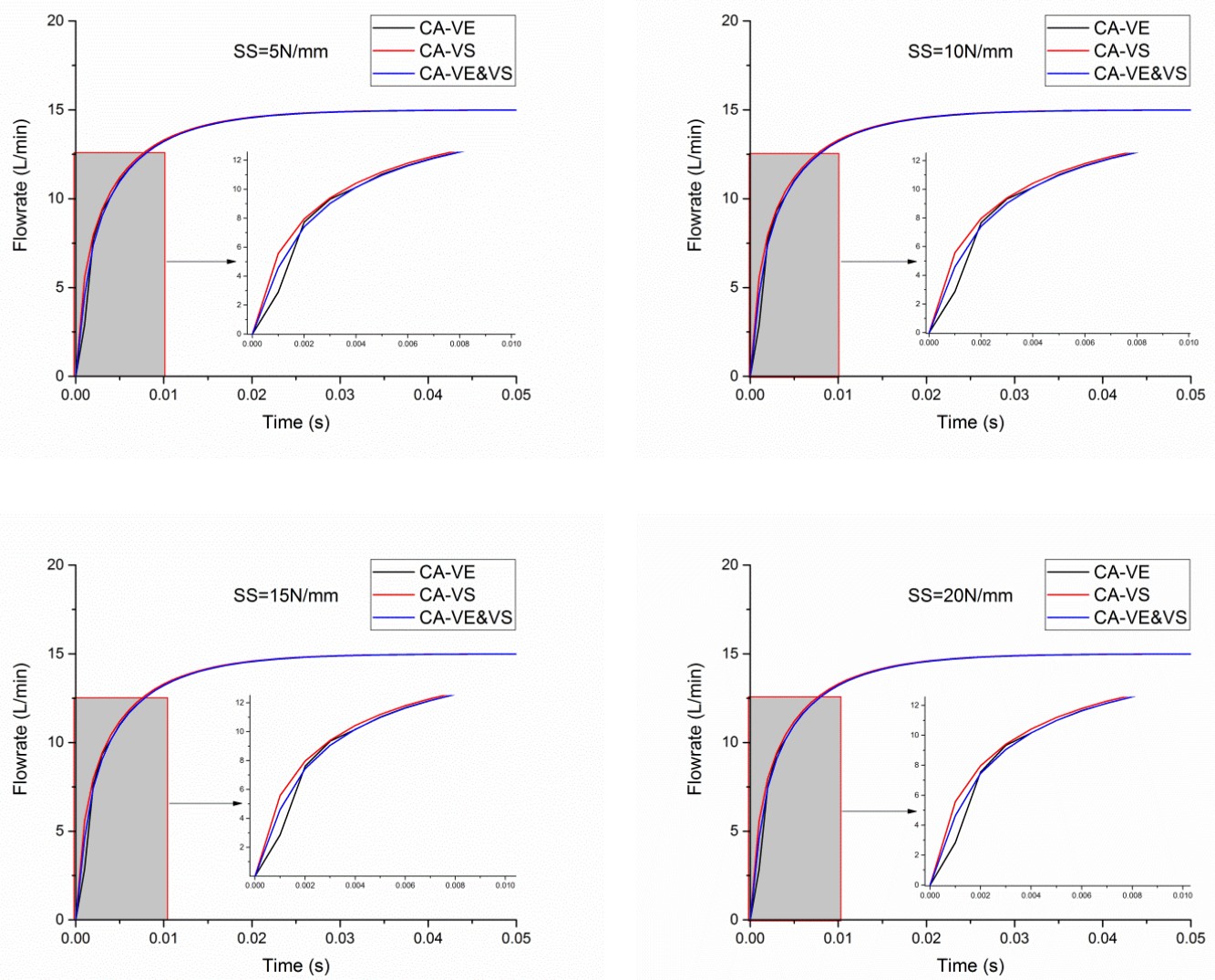

**Figure 12.** The flowrate response of the three kinds of CTDARV: SS = 5 N/mm–20 N/mm.

4.2.5. Effect of CAVE&CAVS on Flowrate Response

Figure 13 shows the effect of CAVE&CAVS (15 degrees–60 degrees) on the flowrate response of the three kinds of CTDARV. It can be clearly seen from Figure 13 that when CAVE&CAVS are 15 degrees–60 degrees, respectively, the three kinds of CTDARV will eventually reach a stable flowrate of 15 L/min. It can be seen from the partial enlarged diagram that when CAVE&CAVS is 15 degrees, CA-VE has relatively large flowrate and CA-VE&VS has relatively small flowrate in the initial stage before 0.001 s. When CAVE&CAVS is 15 degrees, at 0.001 s, the flowrate of CA-VE is 7.8151 L/min, the flowrate of CA-VS is 3.9823 L/min, and the flowrate of CA-VE&VS is 0.5041 L/min. At 0.001 s, with the increase of CAVE&CAVS from 15 degrees to 60 degrees, the flowrate of CA-VS and CA-VE&VS increases. At 0.001 s, the flowrate of CA-VE increases as CAVE&CAVS increases from 45 degrees to 60 degrees. Specifically, when CAVE&CAVS is 15 degrees, 30 degrees, 45 degrees, and 60 degrees; the flowrate of CA-VE is 7.8151 L/min, 5.2183 L/min, 2.8822 L/min, and 3.2828 L/min; the flowrate of CA-VS is 3.9823 L/min, 5.2281 L/min, 5.5783 L/min, and 5.7075 L/min; and the flowrate of CA-VE&VS is 0.5041 L/min, 3.7486 L/min, 4.5983 L/min, and 4.8908 L/min, respectively.

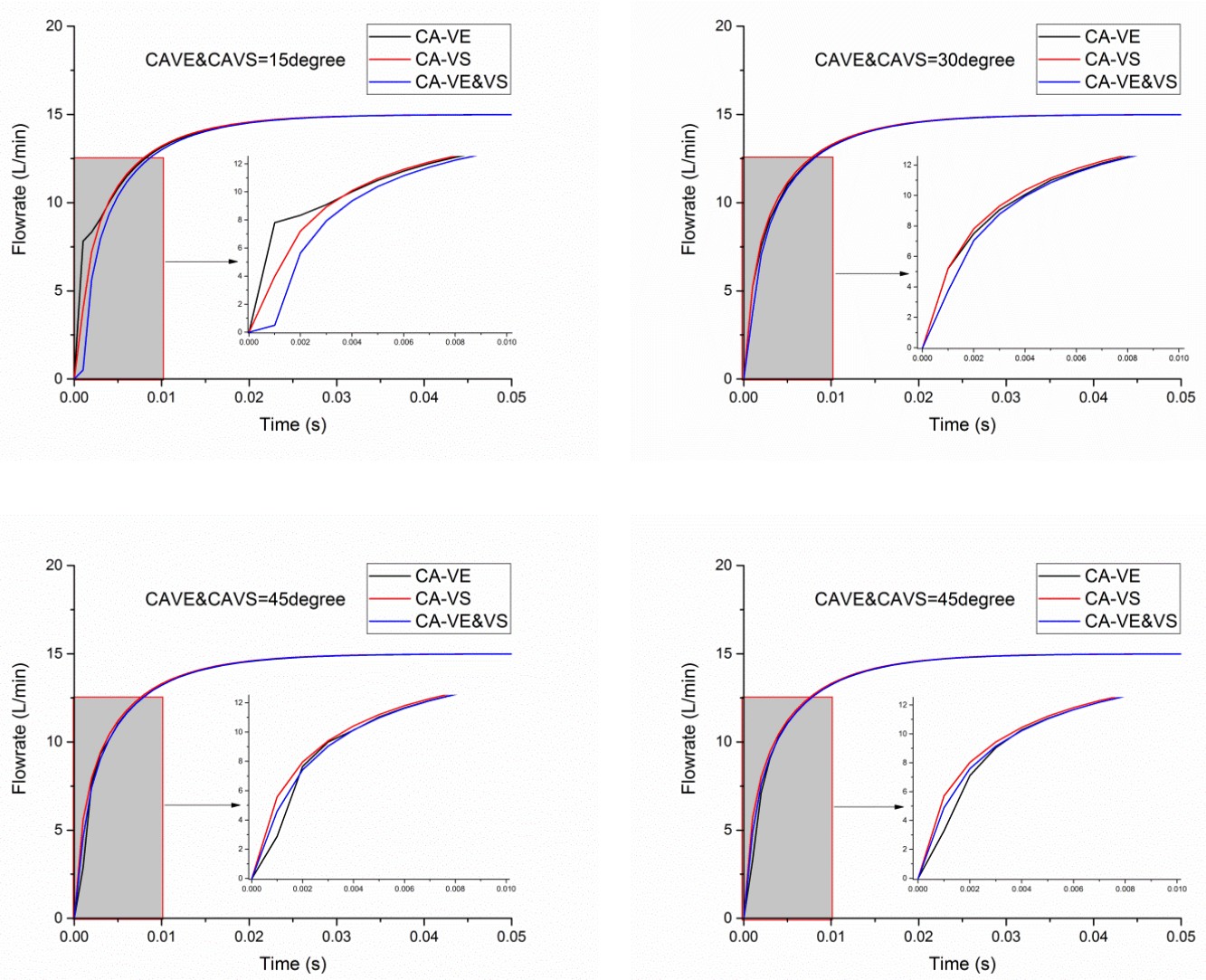

**Figure 13.** The flowrate response of the three kinds of CTDARV: CAVE&CAVS = 15 degrees–60 degrees.

### 4.2.6. Effect of OD on Flowrate Response

Figure 14 shows the effect of OD (0.8 mm–1.4 mm) on the flowrate response of the three kinds of CTDARV. It can be clearly seen from Figure 14 that when CAVE&CAVS are 15 degrees–60 degrees, respectively, the three kinds of CTDARV will eventually reach a stable flowrate of 15 L/min. It can be seen from the partial enlarged diagram that when CAVE&CAVS is 15 degrees, CA-VE has relatively large flowrate and CA-VE&VS has relatively small flowrate in the initial stage before 0.001 s. When CAVE&CAVS is 15 degrees, at 0.001 s, the flowrate of CA-VE is 7.8151 L/min, the flowrate of CA-VS is 3.9823 L/min, and the flowrate of CA-VE&VS is 0.5041 L/min At 0.001 s, with the increase of CAVE&CAVS from 15 degrees to 60 degrees, the flowrate of CA-VS and CA-VE&VS increases At 0.001 s, the flowrate of CA-VE increases as CAVE&CAVS increases from 45 degrees to 60 degrees. Specifically, when CAVE&CAVS is 15 degrees, 30 degrees, 45 degrees, and 60 degrees; the flowrate of CA-VE is 7.8151 L/min, 5.2183 L/min, 2.8822 L/min, and 3.2828 L/min; the flowrate of CA-VS is 3.9823 L/min, 5.2281 L/min, 5.5783 L/min, and 5.7075 L/min; and the flowrate of CA-VE&VS is 0.5041 L/min, 3.7486 L/min, 4.5983 L/min, and 4.8908 L/min, respectively.

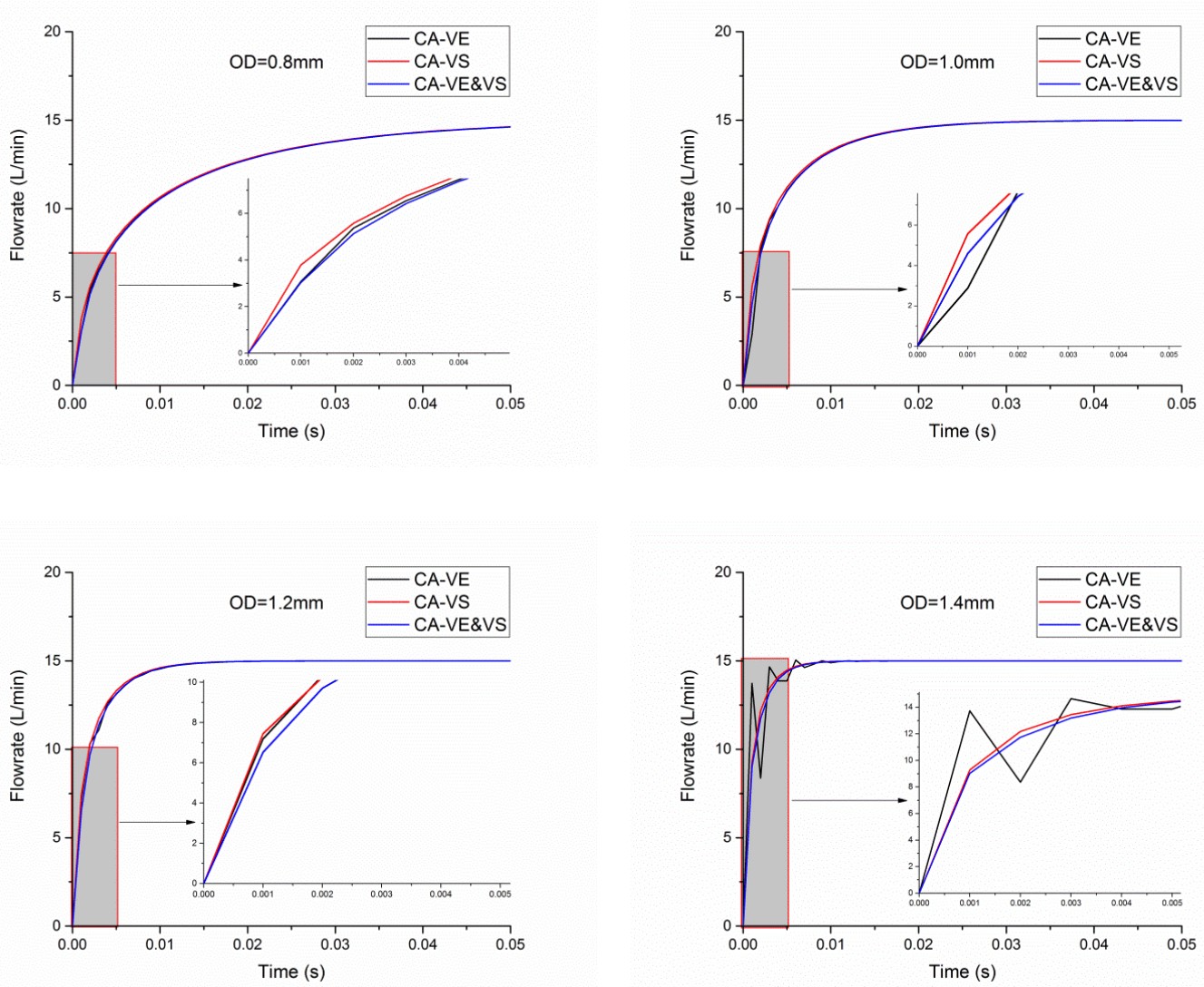

**Figure 14.** The flowrate response of the three kinds of CTDARV: OD = 0.8 mm–1.4 mm.

*4.3. Displacement Performance Comparisons of the Three Kinds of CTDARV*

4.3.1. Effect of VED on Displacement Response

Figure 15 shows the effect of VED (13 mm–16 mm) on the displacement response of the three kinds of CTDARV. It can be seen from Figure 15 that when VED is 13 mm–16 mm, the three kinds of CTDARV will eventually reach their respective stable displacement, of which CA-VE&VS has the highest stable displacement, and CA-VE has the lowest stable displacement. The stable displacement of CA-VE&VS is significantly higher than that of CA-VE and CA-VS. The stable displacement of CA-VE and CA-VS is similar. With the increase of VED from 13 mm to 16 mm, the stable displacement of CA-VE will not change, but the stable displacement of CA-VS and CA-VE&VS will increase, of which the stable displacement of CA-VS will increase slightly, while the stable displacement of CA-VE&VS will increase relatively large. Specifically, when the VED is 13 mm, 14 mm, 15 mm, and 16 mm, respectively, the stable displacement of CA-VE is 0.2587 mm; the stable displacement of CA-VS is 0.2776 mm, 0.2784 mm, 0.2791 mm, and 0.2798 mm, respectively; and the stable pressure displacement of CA-VE&VS is 0.6334 mm, 0.6723 mm, 0.7121 mm, and 0.7527 mm, respectively.

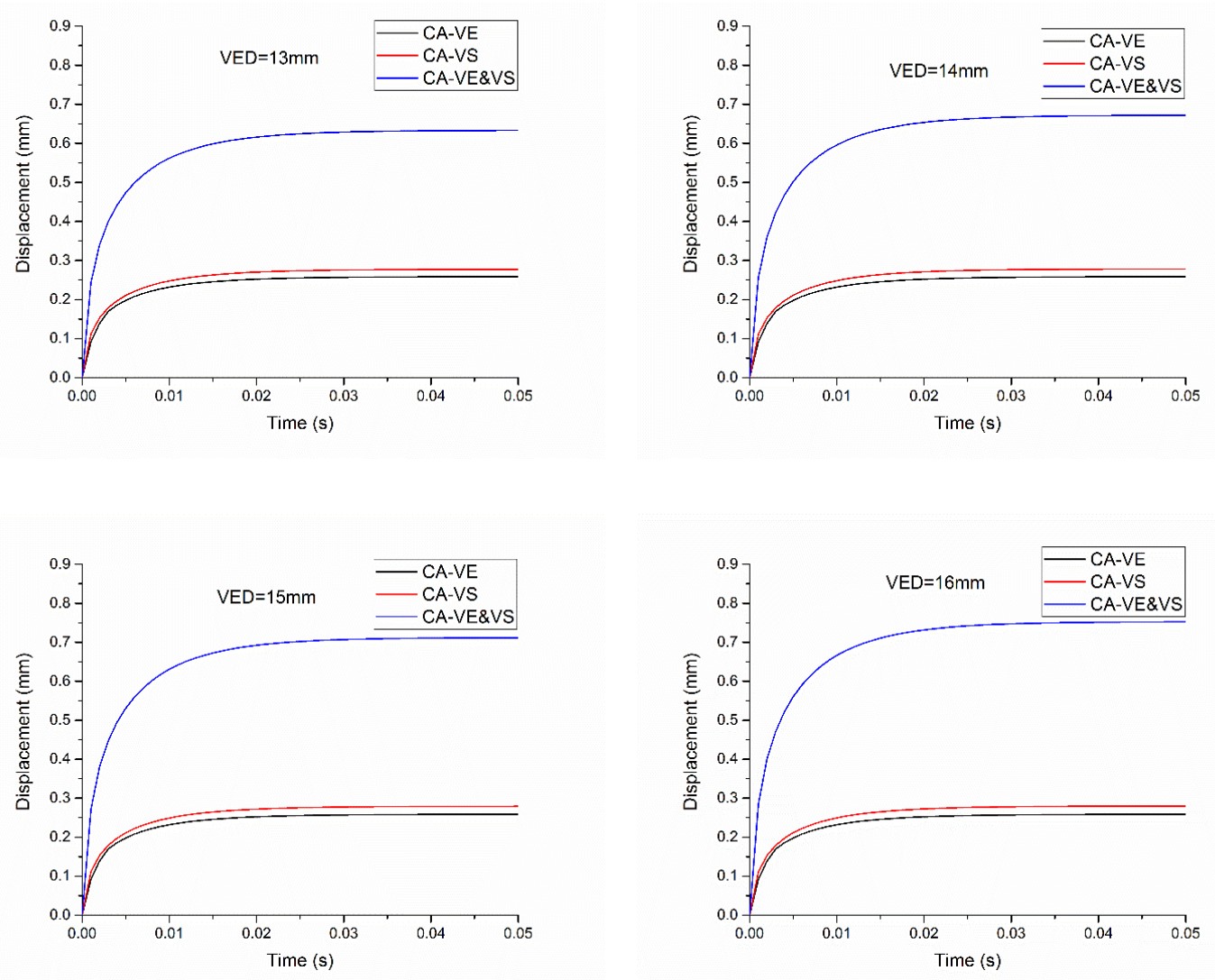

**Figure 15.** The displacement response of the three kinds of CTDARV: VED = 13 mm–16 mm.

4.3.2. Effect of VSD on Displacement Response

Figure 16 shows the effect of VSD (3 mm–6 mm) on the displacement response of the three kinds of CTDARV. It can be clearly seen from Figure 16 that when VSD is 3 mm–6 mm, respectively, the three kinds of CTDARV will eventually reach their respective stable displacement. It is particularly noteworthy that when the VSD is 3 mm, the stable displacement of CA-VS is the highest, reaching 5 mm, and the stable displacement of CA-VE is the lowest, and the stable displacement of CA-VE, CA-VS, and CA-VE&VS differ greatly. When VSD is 4 mm–6 mm, CA-VE&VS has the highest stable displacement and CA-VE has the lowest stable displacement. The stable displacement of CA-VE&VS is significantly higher than that of CA-VE and CA-VS, and the stable displacement of CA-VE and CA-VS has little difference. With the increase of VSD from 3 mm to 6 mm, the stable displacement of CA-VE increases, but the increase is small, while the stable displacement of CA-VE&VS decreases. The stable displacement of CA-VS is 5 mm when VSD is 3 mm. When VSD is 4 mm–6 mm, the stable displacement of CA-VS remains unchanged. Specifically, when the VSD is 3 mm, 4 mm, 5 mm, and 6 mm, respectively, the stable displacement of CA-VE is 0.2393 mm, 0.2513 mm, 0.2587 mm, and 0.2636 mm, respectively; the stable displacement of CA-VS is 5 mm, 0.2791 mm, 0.2791 mm, and 0.2791 mm, respectively; and the stable pressure displacement of CA-VE&VS is 1.5468 mm, 0.9321 mm, 0.7121 mm, and 0.5817 mm, respectively.

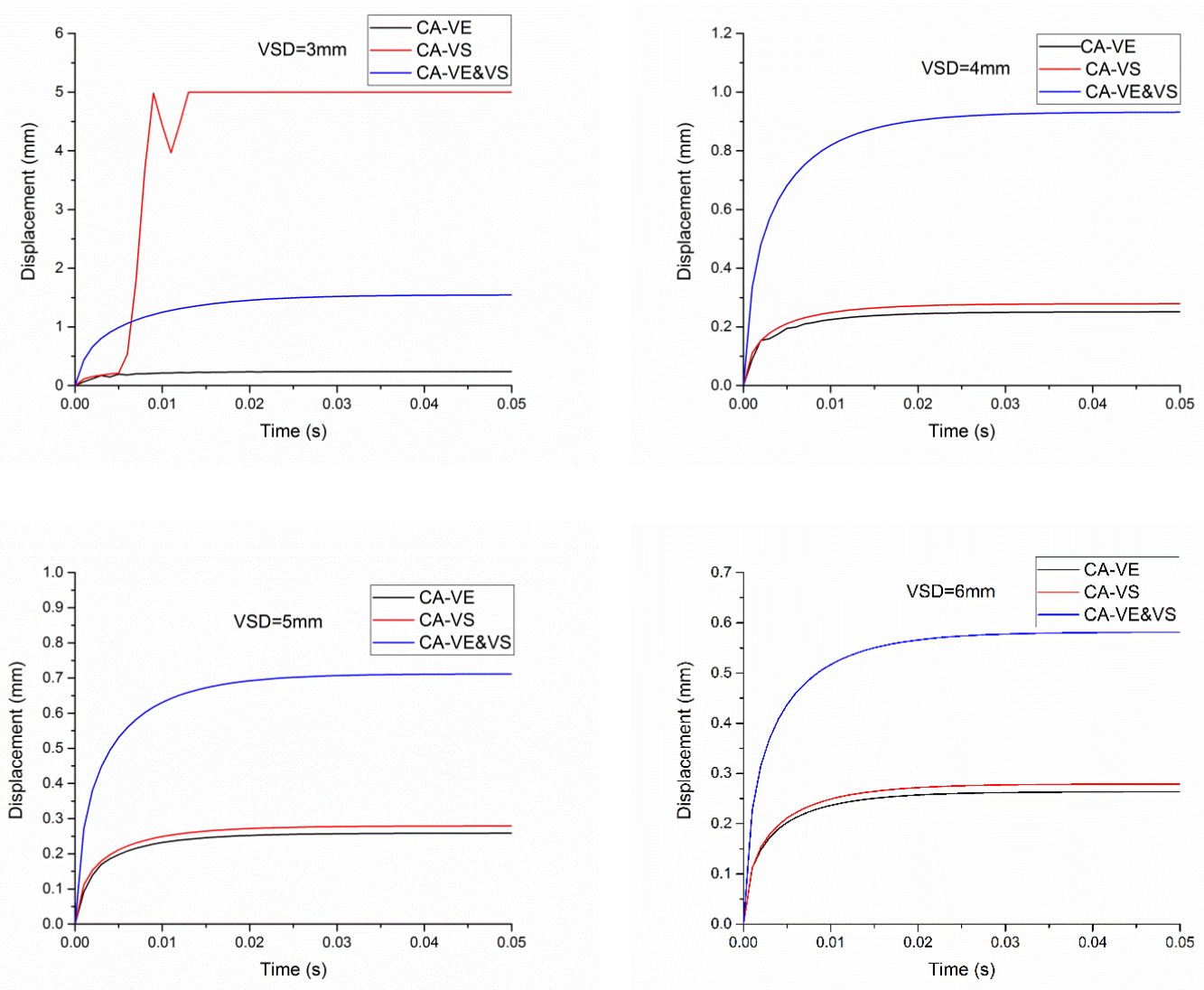

**Figure 16.** The displacement response of the three kinds of CTDARV: VSD = 3 mm–6 mm.

### 4.3.3. Effect of VEM on Displacement Response

Figure 17 shows the effect of VEM (0.01 kg–0.04 kg) on the displacement response of the three kinds of CTDARV. It can be clearly seen from Figure 17 that when the VEM is 0.01 kg–0.04 kg, the three kinds of CTDARV will eventually reach their respective stable displacement. The stable displacement of CA-VE&VS is the highest, and that of CA-VE is the lowest. The stable displacement of CA-VE&VS is significantly higher than that of CA-VE and CA-VS. The difference between the stable displacement of CA-VE and CA-VS is not very large. With the increase of VEM from 0.01 kg to 0.04 kg, the initial displacement of CA-VE oscillates gradually, and the number of oscillations increases gradually, while the displacement of CA-VS and CA-VE&VS does not oscillate. With the increase of VEM, the stable displacement of CA-VE, CA-VS, and CA-VE&VS remain unchanged. Specifically, when the VEM is 0.01 kg, 0.02 kg, 0.03 kg, and 0.04 kg, respectively, the stable displacement of CA-VE is 0.2587 mm, the stable displacement of CA-VS is 0.2791 mm, and the stable pressure displacement of CA-VE&VS is 0.7121 mm.

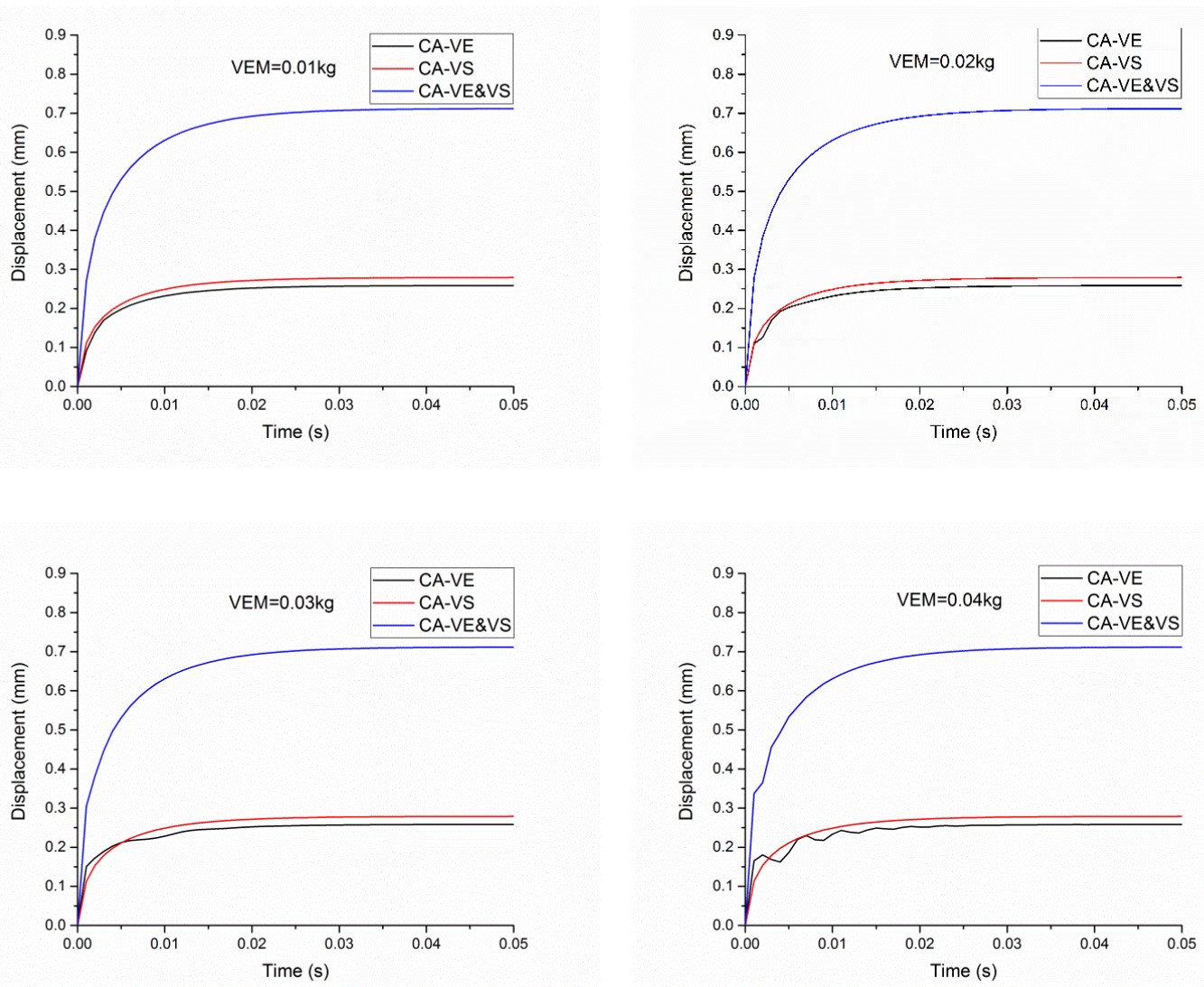

**Figure 17.** The displacement response of the three kinds of CTDARV: VEM = 0.01 kg–0.04 kg.

4.3.4. Effect of SS on Displacement Response

Figure 18 shows the effect of SS (5 N/mm–20 N/mm) on the displacement response of the three kinds of CTDARV. It can be clearly seen from Figure 18 that when SS is 5 N/mm–20 N/mm, the three kinds of CTDARV will eventually reach their respective stable displacement. The stable displacement of CA-VE&VS is the highest, and that of CA-VE is the lowest. The stable displacement of CA-VE&VS is significantly higher than that of CA-VE and CA-VS. The difference between the stable displacement of CA-VE and CA-VS is not very large. With the increase of SS from 5 N/mm to 20 N/mm, the stable displacement of CA-VE, CA-VS and CA-VE&VS decreased, but the decrease was not significant. Specifically, when SS is 5 N/mm, 10 N/mm, 15 N/mm, and 20 N/mm, respectively, the stable displacement of CA-VE is 0.2601 mm, 0.2587 mm, 0.2572 mm, and 0.2559 mm, respectively; the stable displacement of CA-VS is 0.2810 mm, 0.2791 mm, 0.2774 mm, and 0.2756 mm, respectively; and the stable pressure displacement of CA-VE&VS is 0.7246 mm, 0.7121 mm, 0.7005 mm, and 0.6897 mm, respectively.

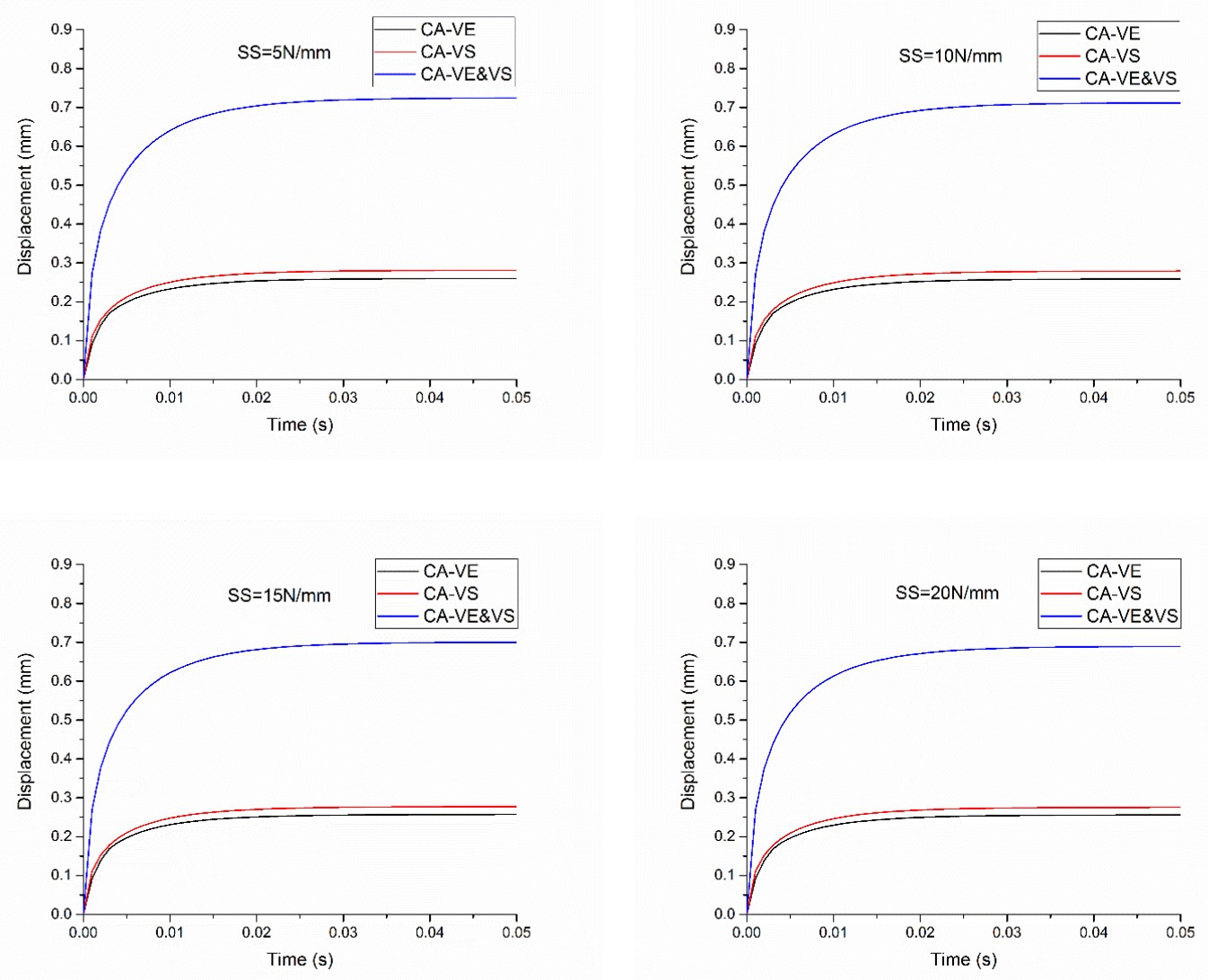

**Figure 18.** The displacement response of the three kinds of CTDARV: SS = 5 N/mm–20 N/mm.

### 4.3.5. Effect of CAVE&CAVS on Displacement Response

Figure 19 shows the effect of CAVE&CAVS (15 degrees–60 degrees) on the displacement response of the three kinds of CTDARV. It can be clearly seen from Figure 19 that when the CAVE&CAVS are 15 degrees–60 degrees, respectively, the three kinds of CT-DARV will eventually reach their respective stable displacement. The stable displacement of CA-VE&VS is the highest, and that of CA-VE is the lowest. The stable displacement of CA-VE&VS is significantly higher than that of CA-VE and CA-VS. The difference between the stable displacement of CA-VE and CA-VS is not very large. With the increase of CAVE&CAVS from 15 degrees to 60 degrees, the stable displacement of CA-VE, CA-VS, and CA-VE&VS decreased, but the decrease was not significant. Specifically, when CAVE&CAVS are 15 degrees, 30 degrees, 45 degrees, and 60 degrees, respectively, the stable displacement of CA-VE is 0.6665 mm, 0.3551 mm, 0.2587 mm, and 0.2188 mm, respectively; the stable displacement of CA-VS is 0.7372 mm, 0.3896 mm, 0.2791 mm and 0.2309 mm, respectively; and the stable compressive displacement of CA-VE&VS is 1.8701 mm, 1.0022 mm, 0.7121 mm, and 0.5782 mm, respectively.

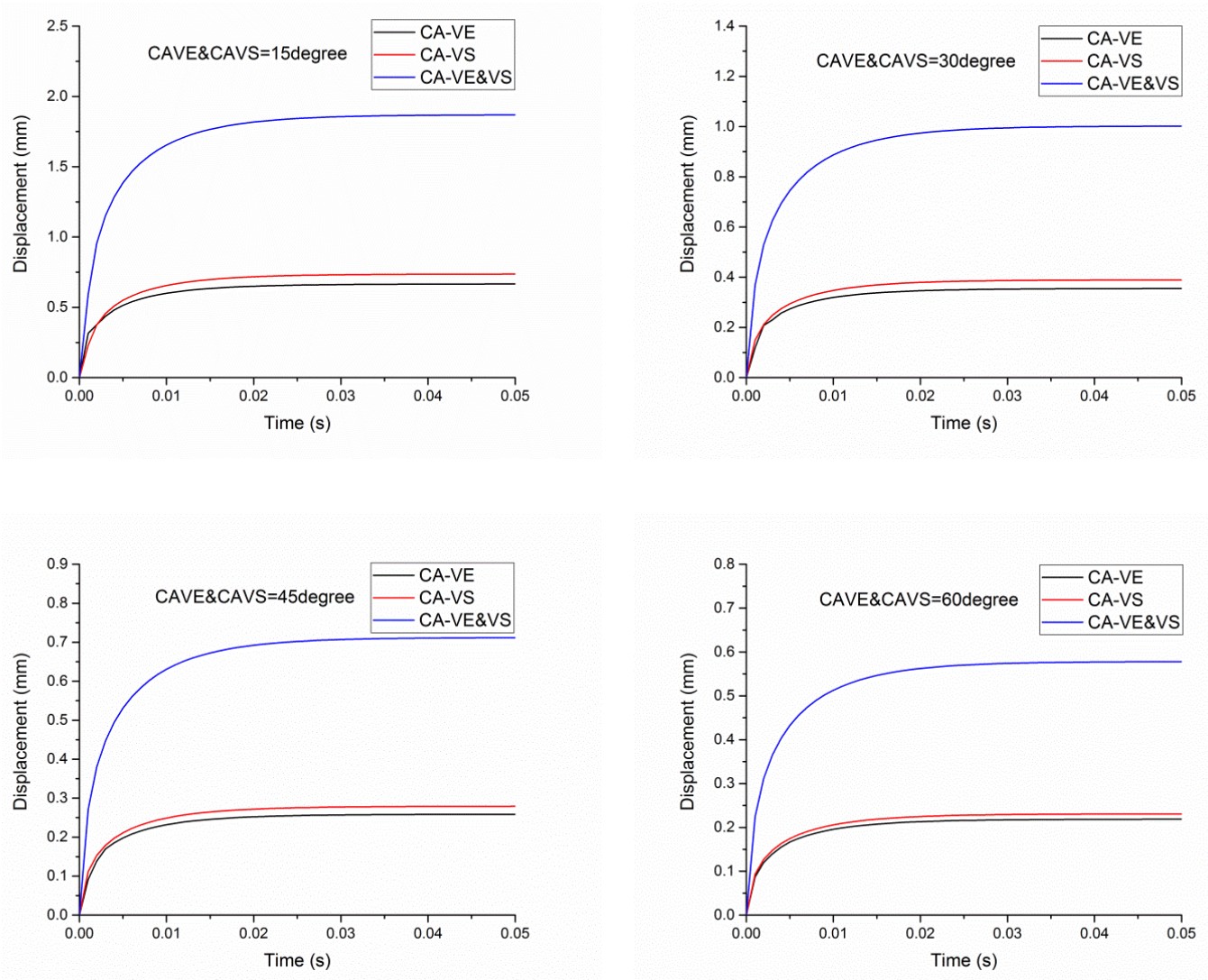

**Figure 19.** The displacement response of the three kinds of CTDARV: CAVE&CAVS = 15 degrees–60 degrees.

### 4.3.6. Effect of OD on Displacement Response

Figure 20 shows the effect of OD (0.8 mm–1.4 mm) on the displacement response of the three kinds of CTDARV. It can be clearly seen from Figure 20 that when the OD is 0.8 mm–1.4 mm, the three kinds of CTDARV will eventually reach their respective stable displacement. The stable displacement of CA-VE&VS is the highest, and that of CA-VE is the lowest. The stable displacement of CA-VE&VS is significantly higher than that of CA-VE and CA-VS. The difference between the stable displacement of CA-VE and CA-VS is not very large. As the OD increases from 0.8 mm to 1.4 mm, the initial displacement of CA-VE oscillates gradually, and the number of oscillations increases gradually, while the displacement of CA-VS and CA-VE&VS does not oscillate. With the OD increasing from 0.8 mm to 1.4 mm, the stable displacement of CA-VE, CA-VS, and CA-VE&VS remained unchanged. Specifically, when the OD is 0.8 mm, 1.0 mm, 1.2 mm, and 1.4 mm, respectively, the stable displacement of CA-VE is 0.2587 mm; the stable displacement of CA-VS is 0.2791 mm; and the stable pressure displacement of CA-VE&VS is 0.7121 mm.

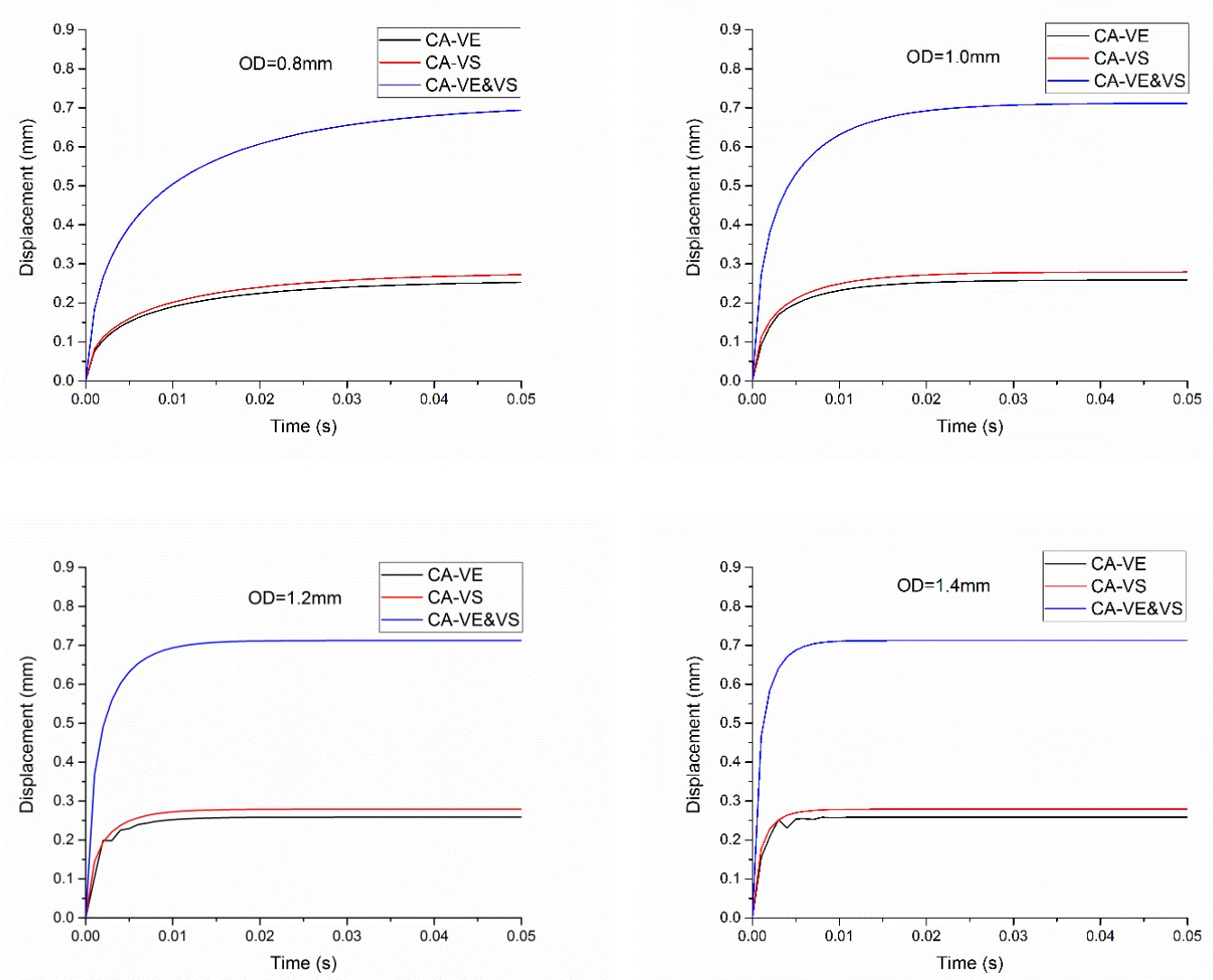

**Figure 20.** The displacement response of the three kinds of CTDARV: OD = 0.8 mm–1.4 mm.

### 4.4. Velocity Performance Comparisons of the Three Kinds of CTDARV

4.4.1. Effect of VED on Velocity Response

Figure 21 shows the effect of VED (13 mm–16 mm) on the velocity response of the three kinds of CTDARV. It can be clearly seen from Figure 21 that when the VED is 13 mm–16 mm, the three kinds of CTDARV will eventually reach the speed of 0 m/s. With the increase of VED from 13 mm to 16 mm, the speed of CA-VE remains unchanged, while the speed of CA-VS and CA-VE&VS increases, but the increase is not significant. The speed of CA-VE, CA-VS, and CA-VE&VS reaches the maximum value in 0.001 s. Specifically, when the VED is 13 mm, 14 mm, 15 mm, and 16 mm, respectively, the speed of CA-VE is 0.2380 m/s; the speed of CA-VS is 0.0555 m/s, 0.0561 m/s, 0.0572 m/s and 0.0591 m/s, respectively; and the speed of CA-VE&VS is 0.1331 m/s, 0.1405 m/s, 0.1487 m/s, and 0.1574 m/s, respectively.

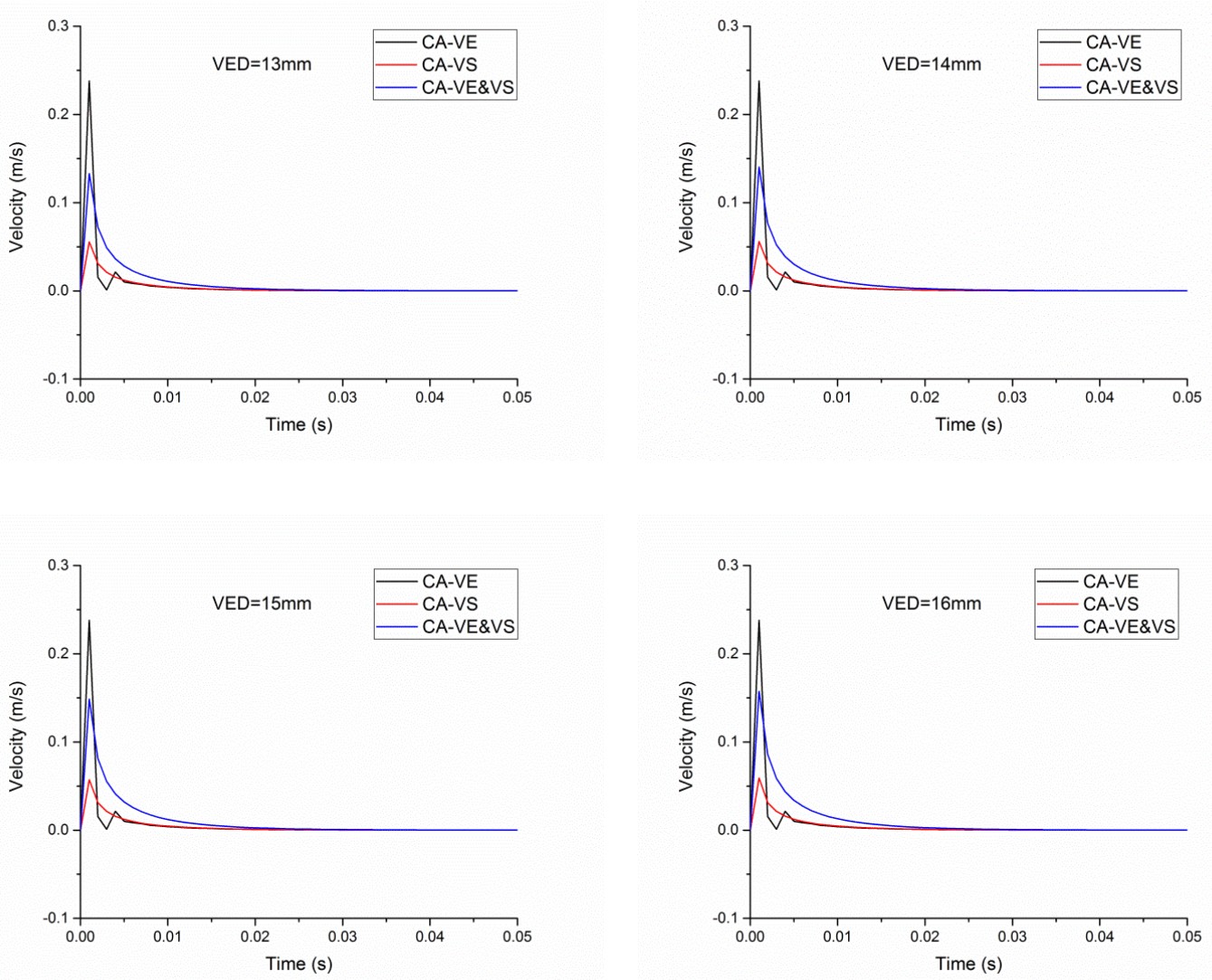

**Figure 21.** The displacement response of the three kinds of CTDARV without orifice: VED = 13 mm–16 mm.

### 4.4.2. Effect of VSD on Velocity Response

Figure 22 shows the effect of VSD (3 mm–6 mm) on the velocity response of the three kinds of CTDARV. It can be clearly seen from Figure 22 that when the VSD is 3 mm–6 mm, the three CTDARVs will eventually reach the speed of 0 m/s. When the VSD is 3 mm, the speed of CA-VS changes sharply, and the maximum speed reaches 1.8760 m/s. As VSD increases from 4 mm to 6 mm, the speed of CA-VS remains unchanged. As VSD increases from 4 mm to 6 mm, the speed of CA-VE and CA-VE&VS decreases. As the VSD increases from 4 mm to 6 mm, the speeds of CA-VE, CA-VS and CA-VE&VS all reach the maximum value in 0.001 s. Specifically, when the VSD are 4 mm, 5 mm, and 6 mm, respectively, in 0.001 s, the speeds of CA-VE are 0.2684 m/s, 0.2380 m/s, and 0.1180 m/s; CA-VS are 0.0572 m/s; and CA-VE&VS are 0.2013 m/s, 0.1487 m/s, and 0.1231 m/s, respectively.

### 4.4.3. Effect of VEM on Velocity Response

Figure 23 shows the effect of VEM (0.01 kg–0.04 kg) on the velocity response of the three kinds of CTDARV. It can be clearly seen from Figure 23 that when the VEM is 0.01 kg–0.04 kg, the three kinds of CTDARV will eventually reach the speed of 0 m/s. With the increase of VEM from 0.01 kg to 0.04 kg, the speed oscillation of CA-VE becomes more and more intense, and the speed of CA-VS and CA-VE&VS changes little.

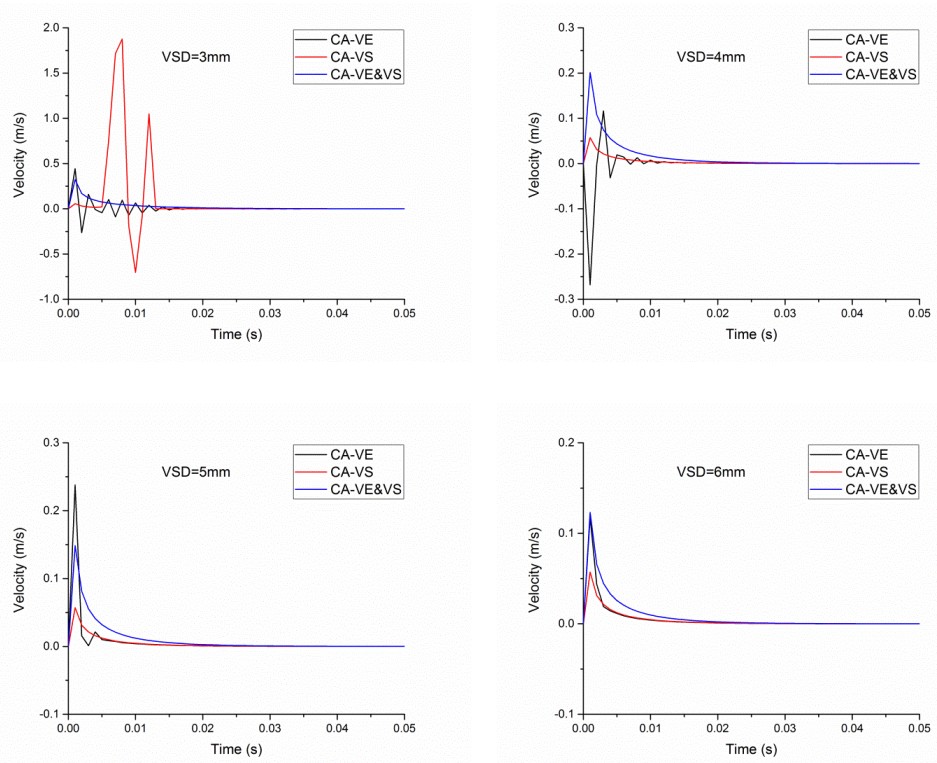

**Figure 22.** The velocity response of the three kinds of CTDARV: VSD = 3 mm–6 mm.

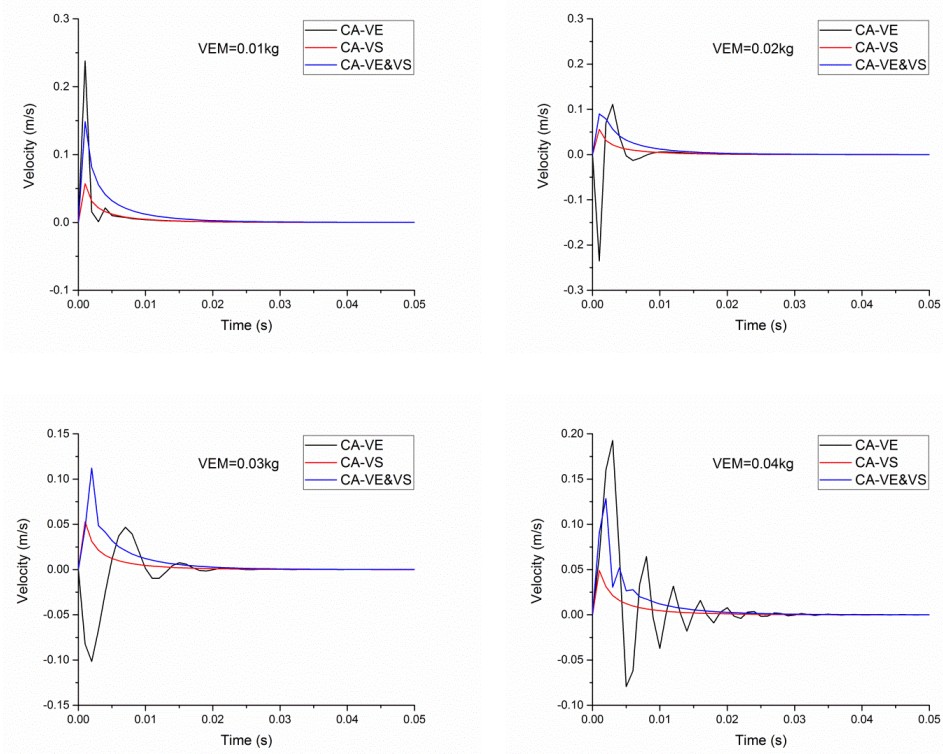

**Figure 23.** The velocity response of the three kinds of CTDARV: VEM = 0.01 kg–0.04 kg.

4.4.4. Effect of SS on Velocity Response

Figure 24 shows the effect of SS (5 N/mm–20 N/mm) on the velocity response of the three kinds of CTDARV. It can be clearly seen from Figure 24 that when SS is 5 N/mm–20 N/mm, the three CTDARVs will eventually reach the speed of 0 m/s and have

similar speed curves. As SS increases from 5 N/mm to 20 N/mm, the speed of CA-VE increases, while the speed of CA-VS and CA-VE&VS decreases, but the increase and decrease are not significant. Specifically, when SS is 5 N/mm, 10 N/mm, 15 N/mm, and 20 N/mm, the speed of CA-VE is 0.2354 m/s, 0.2380 m/s, 0.2403 m/s, and 0.2425 m/s, respectively; the speed of CA-VS is 0.0576 m/s, 0.0572 m/s, 0.0569 m/s, and 0.0566 m/s, respectively; and the speed of CA-VE&VS is 0.1506 m/s, 0.1487 m/s, 0.1468 m/s, and 0.1450 m/s.

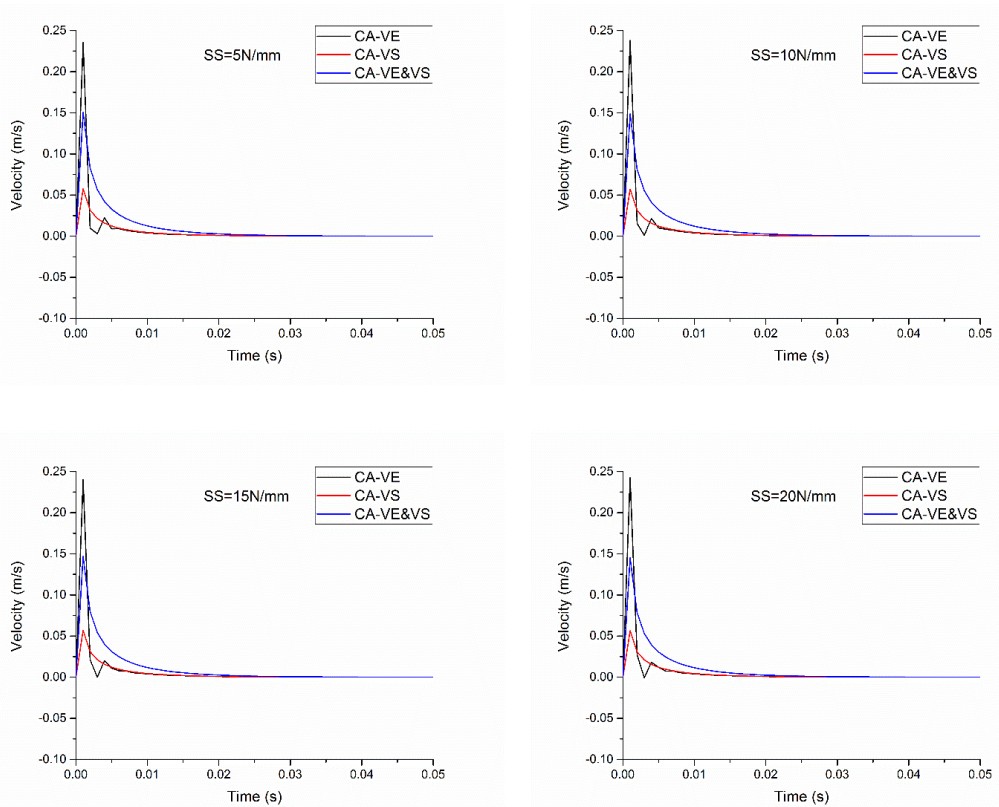

**Figure 24.** The velocity response of the three kinds of CTDARV: SS = 5 N/mm–20 N/mm.

4.4.5. Effect of CAVE&CAVS on Velocity Response

Figure 25 shows the effect of CAVE&CAVS (15 degrees–60 degrees) on the velocity response of the three kinds of CTDARV. It can be clearly seen from Figure 25 that when the CAVE&CAVS are 15 degrees–60 degrees, the three kinds of CTDARV will eventually reach a speed of 0 m/s. At 0.001 s, when CAVE&CAVS is 15 degrees and 30 degrees, the speed of CA-VE is lower than that of CA-VS and CA-VE&VS. At 0.001 s, when CAVE&CAVS is 45 degrees and 60 degrees, the speed of CA-VE is higher than that of CA-VS and CA-VE&VS. Specifically, when CAVE&CAVS is 15 degrees, 30 degrees, 45 degrees, and 60 degrees, the maximum speed of CA-VE is 0.2049 m/s, 0.0410 m/s, 0.2380 m/s, and 0.2242 m/s, and the maximum speed of CA-VS is 0.2077 m/s, 0.0902 m/s, 0.0572 m/s, and 0.0451 m/s, and the maximum speed of CA-VE&VS is 0.5350 m/s, 0.2289 m/s, 0.1487 m/s, and 0.1212 m/s.

4.4.6. Effect of OD on Velocity Response

Figure 26 shows the effect of OD (0.8 mm–1.4 mm) on the velocity response of the three kinds of CTDARV. It can be clearly seen from Figure 26 that when the OD is 0.8 mm–1.4 mm, the three kinds of CTDARV will eventually reach the speed of 0 m/s. With the increase of OD from 0.8 mm to 1.4 mm, the velocity oscillation of CA-VE becomes more and more intense, and the velocity of CA-VS and CA-VE&VS changes little.

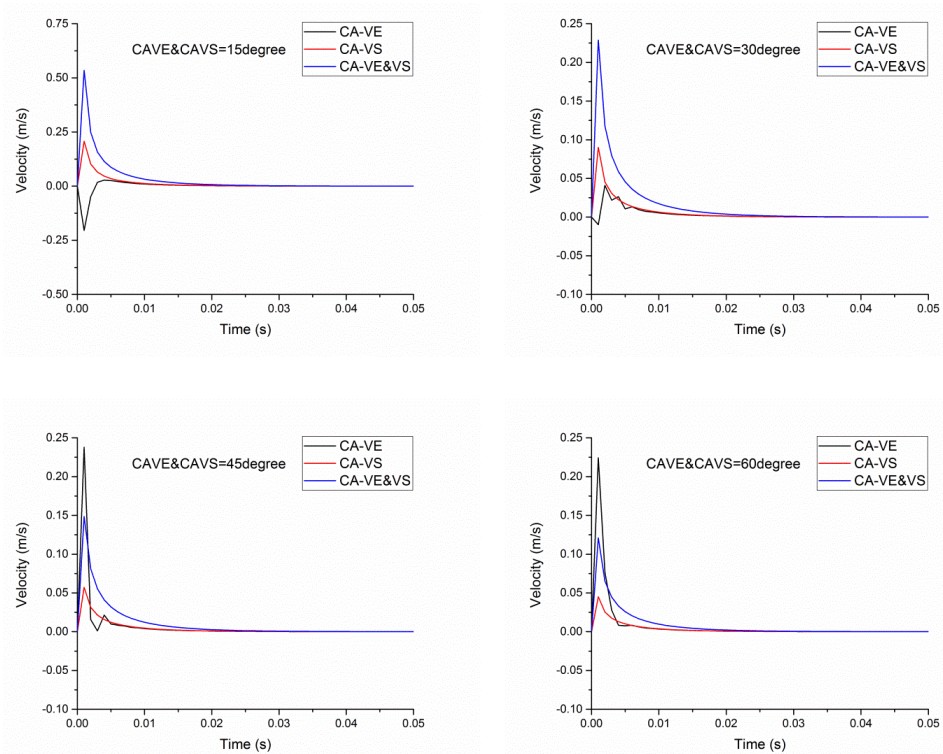

**Figure 25.** The velocity response of the three kinds of CTDARV: CAVE&CAVS = 15 degrees–60 degrees.

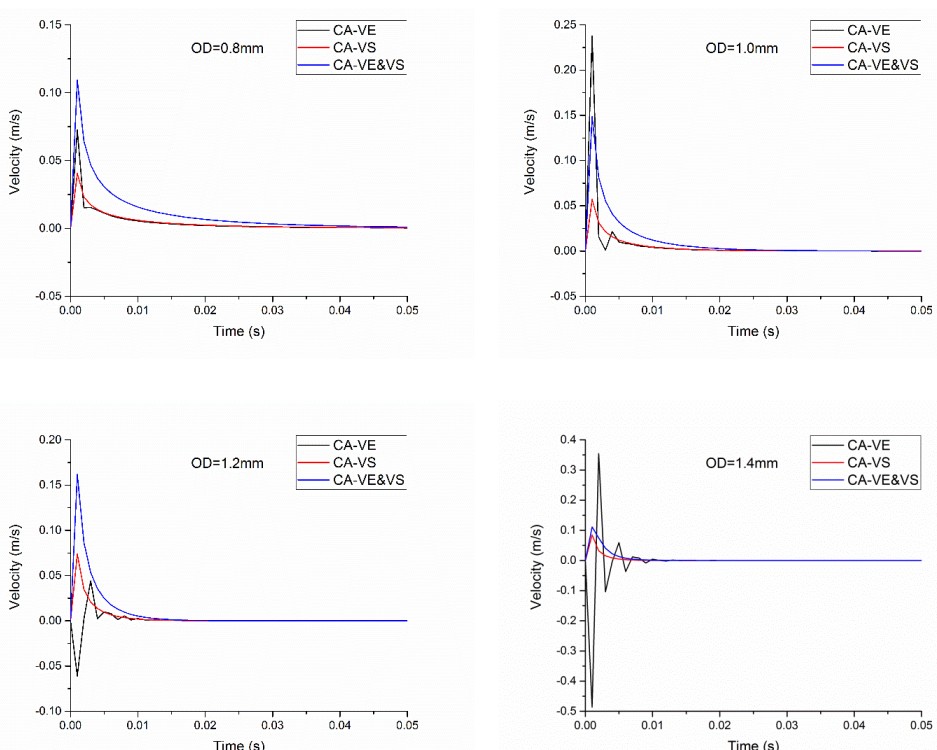

**Figure 26.** The velocity response of the three kinds of CTDARV: OD = 0.8 mm–1.4 mm.

## 5. Conclusions and Future Work

Based on the working principles of the three kinds of CTDARV, the simulation models of the three kinds of CTDARV are established by utilizing AMESIM. The numerical experiments on the three kinds of CTDARV are conducted and the performance comparisons of the three kinds of CTDARV are obtained, and the following conclusions are obtained:

(1) When the values of VED, VSD, VEM, SS, CAVE&CAVS, and OD are the same, the three kinds of CTDARV will eventually reach their respective stable pressure and stable displacement and will eventually reach the same stable flowrate and velocity. When all parameters have the same value, CA-VE has the highest stable pressure, CA-VE&VS has the highest stable displacement, CA-VS has the lowest stable pressure, and CA-VE has the lowest stable displacement. The stable pressure of CA-VE is significantly higher than that of CA-VS and CA-VE&VS. The stable displacement of CA-VE&VS is significantly higher than that of CA-VE and CA-VS, and the stable displacement of CA-VE and CA-VS has little difference.

(2) With the increase of the VED from 13 mm to 16 mm, the stable pressure of CA-VE remains constant, while the stable pressure of CA-VS and CA-VE&VS both decreases. As the VSD increases from 3 mm to 6 mm, the stable pressure of CA-VE and CA-VE&VS decreases, and the stable pressure of CA-VE decreases significantly. The pressure of CA-VS and CA-VE&VS does not fluctuate, and the pressure oscillation of CA-VE decreases gradually. With the increase of VEM from 0.01 kg to 0.04 kg, the stable pressure of CA-VE, CA-VS and CA-VE&VS remained unchanged, the pressure of CA-VS and CA-VE&VS did not fluctuate, and the pressure oscillation of CA-VE gradually increased. With the increase of SS from 5 N/mm to 20 N/mm, the stable pressure of CA-VE, CA-VS, and CA-VE&VS increased, but the increase was very small. With the increase of CAVE&CAVS from 15 degrees to 60 degrees, the stable pressure of CA-VE, CA-VS, and CA-VE&VS decreased, but the decrease was not significant. With the OD increasing from 0.8 mm to 1.4 mm, the stable pressure of CA-VE, CA-VS, and CA-VE&VS remained unchanged, the pressure of CA-VS and CA-VE&VS did not fluctuate, and the pressure oscillation of CA-VE gradually increased.

(3) With the increase of the VED from 13 mm to 16 mm, the stable displacement of CA-VE does not change, while the stable displacement of CA-VS and CA-VE&VS increases. As the VSD increases from 3 mm to 6 mm, the stable displacement of CA-VE increases (though the increase is small), while the stable displacement of CA-VE&VS decreases. When the VSD is 4 mm–6 mm, the stable displacement of CA-VS remains unchanged. With the increase of the VEM from 0.01 kg to 0.04 kg, the stable displacement of CA-VE, CA-VS, and CA-VE&VS remained unchanged. As the SS increases from 5 N/mm to 20 N/mm, the stable displacement of CA-VE, CA-VS, and CA-VE&VS decreases. As CAVE&CAVS increases from 15 degrees to 60 degrees, the stable displacement of CA-VE, CA-VS, and CA-VE&VS decreases. With the increase of OD from 0.8 mm to 1.4 mm, the stable displacement of CA-VE, CA-VS, and CA-VE&VS remained unchanged.

(4) With the increase of the VED from 13 mm to 16 mm, the velocity of CA-VE remained unchanged, while the velocity of CA-VS and CA-VE&VS increased. The velocity of CA-VE, CA-VS, and CA-VE&VS reached the maximum value in 0.001 s. As the VSD increases from 4 mm to 6 mm, the velocity of CA-VS remains unchanged, the velocity of CA-VE and CA-VE&VS decreases, and the velocity of CA-VE, CA-VS, and CA-VE&VS reaches the maximum value at 0.001 s. With the increase of VEM from 0.01 kg to 0.04 kg, the velocity oscillation of CA-VE gradually increases, and the velocity of CA-VS and CA-VE&VS shows little change. As SS increases from 5 N/mm to 20 N/mm, the velocity of CA-VE increases, while the velocity of CA-VS and CA-VE&VS decreases, but the increase and decrease are not significant. When CAVE&CAVS is 15 degrees and 30 degrees, the velocity of CA-VE is lower than that of CA-VS and CA-VE&VS. At 0.001 s, when CAVE&CAVS is 45 degrees and 60 degrees, the velocity of CA-VE is higher than that of CA-VS and CA-VE&VS. With the increase of OD from 0.8 mm to 1.4 mm, the velocity oscillation of CA-VE increases gradually, and the velocity of CA-VS and CA-VE&VS changes little.

In the future, we will manufacture several sets of different series of the three kinds of CTDARV and build a hydraulic test platform of the three kinds of CTDARV to verify the research results on the three kinds of CTDARV achieved in this paper.

**Author Contributions:** Conceptualization, H.L.; methodology, H.L.; validation, Q.Z.; formal analysis, Q.Z.; investigation, Q.Z.; data curation, H.L.; writing—original draft preparation H.L.; writing—review and editing, Q.Z.; supervision, Q.Z.; funding acquisition, H.L. All authors have read and agreed to the published version of the manuscript.

**Funding:** This work is supported by the National Natural Science Foundation of China [grant number 51365008], the Joint Foundation of Department of Science and Technology of Guizhou Province [grant number Qiankehe LH Zi [2015]7658], the Research Foundation of Guizhou Panjiang Coal Power Group Co., Ltd [grant number 701/700878212201], the Science and Technology Special Foundation of Department of Water Resources of Guizhou Province [grant number KT202113]. The authors are grateful to them for their support.

**Data Availability Statement:** The data presented in this study are available on request from the corresponding author.

**Conflicts of Interest:** The authors declare no conflict of interest.

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
