# Peer review of "Numerical Experiments on Performance Comparisons of Conical Type Direct-Acting Relief Valve—With or without Conical Angle in Valve Element and Valve Seat"

_processes, doi:10.3390/pr11061792_

Round 1
Reviewer 1 Report
This is not a scientific paper but a case study on using AMESim. The initial question is indeed interesting (behaviour of valve types) and relevant, yet, the paper lacks any novelty.
- At minimum, the model equations should be provided. How are the valve-seat types different from the modeling point of view? What is the exact model behind the simulations?
- Why is there a difference in the proneness to oscillations? Why is the CA-VE more unstable than the rest? That would be the really interesting question.
- Figure 4. CA-VE with and without orifice: the steady-state pressure difference is the same even though there is an extra orifice. How is this possible? Are the flow rates different?
- For all the computations, the time step is clearly too large.
- Is there any experimental evidence (or literature proof) that the predictions are reliable?
- How are the results predictive for other geometries?
This paper must be rejected in its current form due to the lack of scientific content.
Author Response
Dear Reviewer,
Thank you for your opinions, these comments are very helpful and useful to improve the quality of the manuscript. We have carefully revised our manuscript according to your comments, and further clarified the logic of writing for improving the quality of the manuscript. Now I response your comments with a point by point and highlight the changes in revised manuscript. Full details please see the attachment. We sincerely hope that you find our responses and modifications satisfactory and that the manuscript is now acceptable for publication.
We deeply appreciate your consideration of our manuscript. If you have any queries, please don’t hesitate to contact me at the address below.
Thank you and best regards.
Yours sincerely,
LIU Huiyong
Corresponding author:
Name: LIU Huiyong
E-mail: heartext@163.com

Reviewer 2 Report
Thе authorsr conducts numerical experiments on on three kinds of conical type direct-acting relief valve (CTDARV). Тhe simulation models of CTDARV are established by utilizing AMESIM software. The numerical experiments are conducted and the performance comparisons are obtained.
The authors chose to compare the following species: no conical angle in valve seat, with conical angle in valve seat and no conical angle in valve element and with conical angle in valve element and valve seat.
The simulation parameters include valve body diameter, valve seat diameter, valve element mass, spring stiffness, conical angle of valve element, conical angle of valve seat, orifice diameter. Based on them, a comparison of the obtained results for pressure, flow rate, displacement and velocity was made.
In my opinion, the article is interesting, but does not constitute a thorough scientific study. Some of the results obtained are obvious even without the use of numerical simulation. I suggest that the authors expand the study by including the areas of application of the different options.
Author Response

(The authors gave the same response as above.)

Reviewer 3 Report
At first thank you very much for your interesting article about the behaviour of the configuration of conical angles in valve elements or/and valve seats. The course of action is remaining and comprehensible. Diagrams are clear.
content-relatetd comments:
Only one parameter configuration was examined (regarding to the geometric parameters of the valves and the boundary conditions of the simulation). The results can therefor not be applied generally. For a better understanding it is possible to add this general behaviour in the conclusion with using the underlying regularities.
Future investigations to this valve configuration could be the consideration of temperature dependencies. Because of low strokes of the pistons when used in pilot units the influence of the temperature rises and significantly changes the response behavior.
comments to language and style mistakes:
line 16/205/240/276: maximum
lime 237: headline to next page
In the hope that the comments are approached, the article is well done.
Author Response

(The authors gave the same response as above.)

Reviewer 4 Report
The response of actuators made of various materials in various operating conditions is very important in the conditions of phenomena that take place at high speeds, close to 250Hz. I consider the work as a reference for applications in the field of automation.
Author Response

(The authors gave the same response as above.)
